# Batched Nonparametric Bandits via k-Nearest Neighbor UCB

**Sakshi Arya**                                                    *sxa1351@case.edu*
*Department of Mathematics, Applied Mathematics, and Statistics*
*Case Western Reserve University*

**Reviewed on OpenReview:** *https://openreview.net/forum?id=9gB2EuOPXb*

## Abstract

We study sequential decision-making in batched nonparametric contextual bandits, where actions are selected over a finite horizon divided into a small number of batches. Motivated by constraints in domains such as medicine and marketing, where online feedback is limited, we propose a nonparametric algorithm that combines adaptive k-nearest neighbor (k-NN) regression with the upper confidence bound (UCB) principle. Our method, `BaNk-UCB`, is fully nonparametric, adapts to the context density, and is simple to implement. Unlike prior works relying on parametric or binning-based estimators, `BaNk-UCB` uses local geometry of the contexts to estimate rewards and adaptively balances exploration and exploitation. We provide near-optimal regret guarantees under standard Lipschitz smoothness and margin assumptions, using a theoretically motivated batch schedule that balances regret across batches and achieves minimax-optimal rates. Empirical evaluations on synthetic and real-world datasets demonstrate that `BaNk-UCB` consistently outperforms binning-based baselines.

## 1 Introduction

Many real-world decision-making problems involve using feedback from past interactions to improve future outcomes, a hallmark of adaptive sequential learning. Contextual bandits are a standard framework for modeling these problems, especially in personalized decision-making, where side information helps tailor actions to individuals (Tewari & Murphy, 2017; Li et al., 2010). In this framework, a learner observes a context, selects an action, and receives a reward, aiming to maximize cumulative reward over time through adaptive policy updates.

However, in many practical applications such as clinical trials (Kim et al., 2011; Lai et al., 1983) and marketing campaigns (Schwartz et al., 2017; Mao et al., 2018), adaptivity is limited due to logistical or cost constraints. Decisions are made in batches, and feedback is only received at the end of each batch. This structure permits limited adaptation and renders traditional online bandit algorithms ineffective, motivating new methods tailored for low-adaptivity regimes with few batches.

While parametric bandits have been extended to the batched setting, they often rely on strong modeling assumptions. Nonparametric models offer greater flexibility and robustness (Rigollet & Zeevi, 2010; Qian & Yang, 2016; Reeve et al., 2018a; Zhou et al., 2020), but their use in batched bandits remains limited. Existing nonparametric batched bandit methods, such as `BaSEDB` (Jiang & Ma, 2025), rely on partitioning the context space into bins and treating each bin as a local static bandit instance. While binning-based approaches are effective under structured or uniformly distributed contexts, they rely on fixed spatial partitions that may not adapt well to local variations in context density or geometry. In particular, low-density regions may receive few samples, leading to poor reward estimation and imbalanced exploration across the space. These limitations highlight the need for methods that adapt to the local geometry and data distribution, rather than imposing a fixed spatial discretization, especially in a data-limited setting such as that of batched bandits.

To address this gap, we propose Batched Nonparametric k-nearest neighbor-Upper Confidence Bound (`BaNk-UCB`), a nonparametric algorithm for batched contextual bandits that combines adaptive $k$-nearest neighbor regression with UCB-based exploration. `BaNk-UCB` adapts neighborhood radii to local data density, eliminating the need for manual bin design. Our method adapts neighborhood sizes based on the observed

data distribution, allowing for more flexible and data-driven reward estimation, particularly useful in high-dimensional or heterogeneous settings, even when the global context density is uniformly lower bounded. Under Lipschitz continuity and margin conditions, we prove minimax-optimal regret rates up to logarithmic factors. Empirical results on synthetic and real data show consistent improvements over binning-based methods. Our main contributions are:

- We propose `BaNk-UCB`, a novel nonparametric algorithm for batched contextual bandits that integrates adaptive $k$-nearest neighbor (k-NN) regression with upper confidence bound (UCB) exploration. The method is simple to implement and avoids biases introduced by coarse partitioning of the context space.
- We design a theoretically grounded batch schedule and establish *minimax-optimal regret bound* under standard Lipschitz smoothness and margin conditions. This is, to our knowledge, the first such result for a k-NN-based reward function estimation method in the batched non-parametric setting.
- We highlight how `BaNk-UCB` automatically adapts to the local geometry of the context distribution without requiring explicit modeling assumption, due to the adaptive neighborhood choice in $k$-NN regression.
- We demonstrate through extensive experiments on both synthetic and real-world datasets that `BaNk-UCB` consistently outperforms binning-based baselines, particularly in high-dimensional or heterogeneous contexts.

## 1.1 Related Work

Batched contextual bandits have received growing attention due to their relevance in settings with limited adaptivity, such as clinical trials and campaign-based interventions (Perchet et al., 2016; Gao et al., 2019). Prior work has explored both non-contextual bandits with fixed or adaptive batch schedules (Esfandiari et al., 2021; Kalkanli & Ozgur, 2021; Jin et al., 2021), and contextual bandits, often under parametric assumptions. In particular, linear (Han et al., 2020) and generalized linear models (Ren et al., 2022) have been popular due to their analytical tractability, though such models may fail to generalize when the reward function is nonlinear or misspecified.

Nonparametric bandits have been extensively studied in the fully sequential setting. Early work by Yang & Zhu (2002) employed $\epsilon$-greedy strategies with nonparametric reward estimation. Subsequent methods include the Adaptively Binned Successive Elimination (ABSE) algorithm (Rigollet & Zeevi, 2010; Perchet & Rigollet, 2013), which partitions the context space adaptively and uses elimination-based strategies (Even-Dar et al., 2006). Other approaches include kernel regression methods (Qian & Yang, 2016; Hu et al., 2020), nearest neighbor algorithms (Reeve et al., 2018a; Guan & Jiang, 2018; Zhao et al., 2024), and Gaussian process or kernelized models (Krause & Ong, 2011; Valko et al., 2013; Arya & Sriperumbudur, 2023).

In the batched nonparametric setting, Jiang & Ma (2025) introduced `BaSEDB`, a batched variant of ABSE with dynamic binning and minimax-optimal regret guarantees. Other recent directions include neural network-based estimators (Gu et al., 2024), Lipschitz-constrained models (Feng et al., 2022), and semi-parametric frameworks (Arya & Song, 2025), though each makes different structural assumptions.

Our work departs from these approaches by employing adaptive $k$-nearest neighbor regression to estimate both reward functions and confidence bounds under batch constraints. Unlike binning-based methods which bin the context space into bins of equal width at each batch, `BaNk-UCB` avoids discretization and instead adapts to the local geometry of the context distribution through data-driven neighborhood selection. To our knowledge, this is the first batched nonparametric algorithm based on locally adaptive method like $k$-NN to achieve near-optimal regret guarantees. Empirically, we show that `BaNk-UCB` outperforms the baseline `BaSEDB` across different scenarios, leveraging the well-known ability of $k$-NN to adapt to local geometry of the context space (Kpotufe, 2011).

Our algorithm adapts ideas from the adaptive $k$-NN UCB framework of Zhao et al. (2024), originally designed for online contextual bandits, to the batched setting with delayed feedback, where actions for an entire batch are chosen before any rewards are observed. While both methods use data-dependent $k$-NN regression for reward estimation and confidence bounds, our estimator is batch-aware and tailored to this restricted feedback regime. In particular, our confidence intervals, selection rule, and regret analysis are shaped by the batch structure. We also develop new technical lemmas that clarify and extend key steps from Zhao et al.

(2024) to handle delayed, batch-specific updates, which are crucial for establishing uniform high-probability guarantees and minimax-optimal regret bounds.

## 2 Setup

We consider a batched contextual bandit problem over a finite time horizon $T$, where decisions are grouped into $M$ batches to reflect limited adaptivity. At each round $t \in \{1, \dots, T\}$, a context $X_t \in \mathcal{X} \subset \mathbb{R}^d$ is observed, and the learner selects an action $a_t \in \mathcal{A} = \{1, \dots, K\}$. The learner selects an action $a_t \in \mathcal{A}$ based on $X_t$ and receives a noisy reward:

$$Y_t = f_{a_t}(X_t) + \epsilon_t, \tag{1}$$

where $f_a(x)$ is an unknown mean reward function for $a \in \mathcal{A}$ and $x \in \mathcal{X}$. The model noise is given by $\epsilon_t$. We make the following assumptions on the noise and context space.

**Assumption 1** (Sub-Gaussian noise). *We assume that the noise terms $\{\epsilon_t\}_{t=1}^T$ are independent and $\sigma^2$-sub-Gaussian; that is, for all $\lambda \in \mathbb{R}$ and all $t$,*

$$\mathbb{E}\left[e^{\lambda \epsilon_t}\right] \leq e^{\frac{1}{2}\lambda^2 \sigma^2}. \tag{2}$$

**Assumption 2** (Bounded context density). *The context vectors $X_t$ are drawn i.i.d. from a distribution with density $p_X$, which is supported on $\mathcal{X} \subset \mathbb{R}^d$. We assume that $p_X(x) \geq \underline{c}$ for some $\underline{c} > 0$.*

Unlike many existing nonparametric bandit algorithms, such as `ABSE` (Perchet & Rigollet, 2013) and its batched variant `BaSEDB` (Jiang & Ma, 2025), which rely on uniform binning of the context space (typically assuming a hypercube domain such as $[0,1]^d$), our proposed method accommodates *arbitrary bounded domains* $\mathcal{X} \subset \mathbb{R}^d$ with densities bounded away from zero. For instance, $\mathcal{X}$ may be a spherical or manifold-shaped domain where uniform partitioning is either ill-defined or computationally inefficient. In contrast to binning-based methods that depend on rigid geometric structure to define partitions and control coverage, our $k$-NN based approach naturally adapts to the local data geometry, eliminating the need for explicit grid design and enabling applicability to more general, heterogeneous settings. This adaptivity is particularly crucial in *data-limited regimes* such as batched bandits, where learning can only occur at a small number of decision points.

A *policy* $\pi_t : \mathcal{X} \to \mathcal{A}$ for $t = 1, \dots, T$ determines an action $a_t \in \mathcal{A}$ at $t$. Based on the chosen action $a_t$, a reward $Y_t$ is obtained. In the sequential setting without batch constraints, the policy $\pi_t$ can depend on all the observations $(X_s, Y_s)$ for $s < t$. In contrast, in a batched setting with $M$ batches, where $0 = t_0 < t_1 < \cdots < t_{M-1} < t_M = T$, for $t \in [t_i, t_{i+1})$, the policy $\pi_t$ can depend on observations from the previous batches, but not on any observations within the same batch. In other words, policy updates can occur only at the predetermined batch boundaries $t_1, \dots, t_M$. This reflects the constraint that feedback is only revealed at the end of each batch.

Let $\mathcal{G} = \{t_0, t_1, \dots, t_M\}$ represent a partition of time $\{0, 1, \dots, T\}$ into $M$ intervals, and $\pi = (\pi_t)_{t=1}^T$ be the sequence of policies applied at each time step. We define the cumulative regret of a policy $\pi$ as:

$$R_T(\pi) = \sum_{t=1}^T f_*(X_t) - f_{(\pi_t(X_t))}(X_t), \tag{3}$$

where $f_*(x) = \max_{a \in \mathcal{A}} f_a(x)$ is the expected reward from the optimal choice of arms given a context $x$. The expected cumulative regret, denoted by $\mathcal{R}_T(\pi) = \mathbb{E}[R_T(\pi)]$, averages over the randomness in $(X_t, a_t)$ under the policy $\pi$. We focus on designing batched policies that minimize $\mathcal{R}_T(\pi)$. The cumulative regret serves as a pivotal metric, quantifying the difference between the cumulative reward attained by $\pi$ and that achieved by an optimal policy, assuming perfect foreknowledge of the optimal action at each time step.

We make the following assumptions on the reward functions.

**Assumption 3** (Lipschitz Smoothness). *We assume that the link function $f_a : \mathbb{R}^d \to \mathbb{R}$ for each arm is Lipschitz smooth, that is, there exists $L > 0$ such that for $a \in \mathcal{A}$,*

$$|f_a(x) - f_a(x')| \leq L\|x - x'\|,$$

*holds for $x, x' \in \mathcal{X}$.*

**Assumption 4** (Margin)**.** *For some $0 < \alpha \leq d$ and for all $a \in \mathcal{A}$, there exists a $\delta_0 \in (0,1)$ and $D_\alpha > 0$ such that*

$$\mathbb{P}_X(0 < f_*(X) - f_a(X) \leq \delta) \leq D_\alpha \delta^\alpha,$$

*holds for all $\delta \in [0, \delta_0]$.*

For $K = 2$, our margin condition reduces exactly to that of Rigollet & Zeevi (2010), which bounds the probability mass where the two arms are nearly indistinguishable. For $K > 2$, we extend this condition to hold for each suboptimal arm relative to the best arm. The margin condition implies that the regions where the reward gap is small, i.e., where it is hard to distinguish the best arm are not too large. The exponent $\alpha$ controls the rate at which the measure of such regions shrinks as $\delta \to 0$. When $\alpha$ is small, suboptimal arms can be frequently indistinguishable from the best arm, leading to slower learning; larger $\alpha$ implies faster decay and enables faster convergence.

**Remark 1.** *Throughout this paper, we assume that $\alpha \leq d$. This follows the result of Perchet & Rigollet (2013), who show that under Hölder smoothness with exponent $\beta$, nontrivial oracle policies exist only when $\alpha\beta \leq d$, while the oracle becomes trivial when $\alpha\beta > d$. In the trivial regime, the optimal policy $\pi^\star$ always pulls the same arm at every time step, almost surely under $\mathbb{P}_X$. In such cases, contextual information becomes irrelevant for decision-making. In our Lipschitz setting ($\beta = 1$), this reduces to the condition $\alpha \leq d$, ensuring that contexts play a meaningful role in learning and regret minimization.*

The margin condition plays a crucial role in determining the minimax rate of regret in nonparametric bandit problems, similar to its role in classification (Mammen & Tsybakov, 1999; Tsybakov & Audibert, 2007).

**Notation:** We use $\|\cdot\|$ to denote the Euclidean norm in $\mathbb{R}^d$. We denote $B(x, r)$ to denote a Euclidean ball with center $x \in \mathbb{R}^d$ and radius $r$. We denote $\lesssim$ and $\gtrsim$ to denote inequalities upto constants. The notation $f(n) = \Theta(g(n))$ indicates an asymptotic tight bound. Formally, there exist positive constants $c_1, c_2$ and $n_0$ such that for all $n \geq n_0$, $c_1 \cdot g(n) \leq f(n) \leq c_2 \cdot g(n)$. The notation $\tilde{O}(g(n))$ denotes an asymptotic upper bound up to logarithmic factors. For $a, b \in \mathbb{R}$, $a \vee b$ denotes the maximum of $a$ and $b$, and $a \wedge b$ denotes minimum of $a$ and $b$. For any batch $m$, let $\mathcal{F}_{t_m}$ be the filtration encoding the history up to batch $m$.

## 3 Batched Nonparametric k-Nearest Neighbor-UCB (`BaNk-UCB`) Algorithm

Recall that in the batched bandits setting, the decision at time $t$ in batch $m$ only depends on the information observed up to the end of the $(m-1)^{\text{th}}$ batch. We propose `BaNk-UCB` (**B**atched **N**onparametric **k**-Nearest Neighbors **U**pper **C**onfidence **B**ound), described in Algorithm 1. The algorithm is based on an *adaptive k-nearest neighbor* policy that tunes the neighborhood size $k$ based on the local sub-optimality gap (margin) and context density. This approach extends the adaptive $k$-NN UCB algorithm of Zhao et al. (2024) for the online setting to the batched nonparametric bandit setting. Let us first define some useful notation. For $x \in \mathcal{X}$ and some fixed $k \leq t_{m-1}$, let $N_{t_{m-1},k}(x, a)$ be the set of $k$ nearest neighbors of $x$ where arm $a$ was chosen, i.e.,

$$N_{t_{m-1},k}(x, a) := \{s \leq t_{m-1} : a_s = a \text{ and } X_s \text{ is among the } k \text{ nearest to } x\}. \tag{4}$$

For simplicity, we denote $N_{t,k}(x, a) \equiv N_{t_{m-1},k}(x, a)$ for all times $t$ within the batch interval $(t_{m-1}, t_m]$. Then we define for $t \in (t_{m-1}, t_m]$,

$$d_{a,t,k}(x) = \max_{s \in N_{t_{m-1},k}(x,a)} \|X_s - x\|, \tag{5}$$

to be the radius of the $k$-NN ball around $x$ for arm $a$. We adaptively select the number of neighbors, denoted $k_{a,t}(x)$, based solely on observations available up to the end of batch $(m-1)$ and specifically associated with arm $a$. This $k_{a,t}$ is then used in the proposed `BaNk-UCB` algorithm as described in Algorithm 1:

$$k_{a,t}(x) = \max\left\{ j \mid L d_{a,t,j}(x) \leq \sqrt{\frac{\ln t_{m-1}}{j}} \right\}. \tag{6}$$

Note that, the left hand side $L d_{a,t,j}(x)$ controls the bias in the estimation of $f_a$ and the right-hand side $\sqrt{\ln t_{m-1}/j}$ controls the variance in the estimation, i.e., it ensures that we use large $k$ if previous samples

are relatively dense around $X_t$, and vice versa. The adaptive selection of $k$ in equation 6 requires that the nearest observed context be sufficiently close. Specifically, we enforce $Ld_{a,t,1}(X_t) \leq \sqrt{\ln t_{m-1}}$; otherwise, reliable estimation is not feasible, and we conservatively set the UCB to infinity: $\hat{f}_{a,t}(x) = \infty$. Otherwise, for $t \in (t_{m-1}, t_m]$, we calculate the upper confidence bound (UCB) as follows:

$$\hat{f}_{a,t}(x) = \frac{1}{k_{a,t}(x)} \sum_{s \in N_{t_{m-1}}(x,a)} Y_s + \xi_{a,t}(x) + Ld_{a,t,k}(x), \tag{7}$$

where $d_{a,t}$ is as defined in equation 5 and $\xi_{a,t}$ is defined as:

$$\xi_{a,t}(x) = \sqrt{\frac{2\sigma^2}{k_{a,t}(x)} \ln\left(dt_{m-1}^{2d+3}|\mathcal{A}|\right)}. \tag{8}$$

---

**Algorithm 1** BaNk-UCB for Batched Nonparametric Bandits

1: Input: Partition $t_0, t_1, \ldots, t_M$, with $t_0 = 0$ and $t_M = T$.
2: **for** $m = 1, \ldots, M$ **do**
3:     **for** $t = t_{m-1} + 1, \ldots, t_m$ **do**
4:         Receive context $X_t$;
5:         **for** $a \in \mathcal{A}$ **do**
6:             **if** $Ld_{a,t,1}(X_t) > \sqrt{\ln t_{m-1}}$ **then**
7:                 Set $\hat{f}_{a,t}(X_t) \leftarrow +\infty$;
8:             **else**
9:                 Compute $k_{a,t}(X_t)$ according to equation 6;
10:                Compute $\hat{f}_{a,t}(X_t)$ according to equation 7;
11:             **end if**
12:         **end for**
13:         Choose action $a_t = \arg\max_{a \in \mathcal{A}} \hat{f}_{a,t}(X_t)$;
14:         Pull arm $a_t$;
15:     **end for**
16:     Observe rewards $\{Y_t, t \in t_{m-1} + 1, \ldots, t_m\}$;
17: **end for**

---

Here, $\xi_{a,t}(x)$ provides a high-probability bound for stochastic noise of the nearest-neighbor averaging, while $Ld_{a,t}(x)$ controls the estimation bias from finite-sample approximation. Both terms depend explicitly on prior-batch data, highlighting the critical role batch design plays in balancing estimation accuracy and cumulative regret. Finally, the algorithm selects arm $a_t$ with the maximum UCB value,

$$a_t = \arg\max_{a \in \mathcal{A}} \hat{f}_{a,t}(X_t). \tag{9}$$

Note that in equation 9, ties are broken arbitrarily at each time step $t$.

The adaptive choice of $k_{a,t}(x)$ in equation 6 simultaneously balances the bias-variance and exploration-exploitation trade-offs in estimating $f_a$. Specifically, the bias-variance trade-off is managed by selecting a larger $k$ when previously observed contexts are densely sampled around $X_t$, thereby reducing variance, and choosing a smaller $k$ otherwise, controlling bias. Moreover, due to the Lipschitz smoothness assumption, contexts with larger optimality gaps ($f_*(x) - f_a(x)$) naturally correspond to larger radii $d_{a,t,j}(x)$, leading to smaller chosen values of $k$ and promoting targeted exploration in regions with high uncertainty.

Note that, a key distinction from Jiang & Ma (2025) lies in how structural assumptions influence the algorithm. In their method, the design of the partition grid explicitly depends on the unknown margin parameter $\alpha$. *In contrast, our adaptive choice of $k$ in equation 6 in the k-NN estimator does not require direct knowledge of $\alpha$, making our approach more robust to unknown margin condition.*

**Remark 2** (Tuning hyperparameters)**.** *Assuming a known sub-Gaussian variance proxy $\sigma^2$ in Assumption 1 is standard in the literature. For example, Reeve et al. (2018b) assume unit variance, and Perchet &*

*Rigollet (2013) assume bounded rewards, which imply sub-Gaussian tails. We make this dependence explicit for transparency in regret scaling. In practice, since $\sigma^2$ enters the UCB width in equation 8, it can be conservatively overestimated, tuned, or estimated from early residuals. Similarly, the Lipschitz constant $L$, which also appears in the confidence bound, can be treated as a tunable hyperparameter or heuristically chosen based on preliminary runs.*

## 4 Minimax Analysis on the Expected Regret

In this section, we demonstrate that the BaNk-UCB algorithm achieves a minimax optimal rate on the expected cumulative regret under an appropriately designed partition of grid points. Specifically, the rate matches known minimax lower bound up to logarithmic factors. First we describe the choice of the batch grid points and then state the upper and lower bounds on the expected regret.

### 4.1 Batch sizes

The choice of batch sizes plays a crucial role in the performance of the batched bandit algorithms. We partition the time horizon into $M$ batches, denoted by grid points $\mathcal{G} = \{t_1, t_2, \ldots, t_M\}$, with $t_0 = 0$. The special case $M = T$ recovers the fully sequential bandit setting, where policy updates occur at every step. Conversely, smaller $M$ imposes fewer policy updates, introducing a trade-off between computational/operational complexity and regret accumulation. A key challenge in the batched setting is selecting the grid $\mathcal{G}$. Intuitively, to minimize total regret, no single batch should dominate the cumulative error, suggesting that the grid should balance regret across batches. If one batch incurs higher regret, reassigning time steps can improve the overall rate. This motivates a grid choice that equalizes regret across batches, up to order in T and d, as we formalize below. We choose:

$$t_1 = ad, \quad t_m = \lfloor at_{m-1}^{\gamma} \rfloor, \tag{10}$$

where $\gamma = \frac{1+\alpha}{2+d}$ and $a = \Theta(T^{\frac{1-\gamma}{1-\gamma^M}})$ is chosen so that $t_M = T$.

### 4.2 Regret bounds

In order to establish the regret rates, we first define the batch-wise expected sample density, motivated by the formulation of Zhao et al. (2024). Let $p_a^{(m)} : \mathcal{X} \to \mathbb{R}$ is defined such that for all $A \subseteq \mathcal{X}$,

$$\mathbb{E}\left[\sum_{t=t_{m-1}}^{t_m} 1(X_t \in A, a_t = a)\right] = \int_A p_a^{(m)}(x)dx. \tag{11}$$

First let's consider the cumulative regret relate it to the batch-wise expected sample density.

**Lemma 1.** *The expected cumulative regret in equation 3 is given by* $R_T(\pi) = \sum_{a \in \mathcal{A}} \sum_{m=1}^{M} R_a^{(m)}(\pi)$*, where* $R_a^{(m)}(\pi)$ *is defined as:*

$$R_a^{(m)}(\pi) = \int_{\mathcal{X}} (f_*(x) - f_a(x))p_a^{(m)}(x)dx. \tag{12}$$

*Proof.* Consider,

$$R_T(\pi) = \mathbb{E}\left[\sum_{t=1}^{T}(f_*(X_t) - f_{a_t}(X_t))\right]$$

$$= \mathbb{E}\left[\sum_{m=1}^{M}\sum_{t=t_{m-1}}^{t_m}(f_*(X_t) - f_{a_t}(X_t))\right]$$

$$= \sum_{a \in \mathcal{A}}\sum_{m=1}^{M}\mathbb{E}\left[\sum_{t=t_{m-1}}^{t_m}(f_*(X_t) - f_{a_t}(X_t))1(a_t = a)\right]$$

$$= \sum_{a \in \mathcal{A}}\sum_{m=1}^{M}\int_{\mathcal{X}}(f_*(x) - f_{a_t}(x))\, p_a^{(m)}(x)dx.$$

$\square$

Note that, in the third equation we rewrite the expectation over $a_t \sim \pi(\cdot \mid X_t)$ as a deterministic sum over all arms $a \in \mathcal{A}$, using the indicator $1(a_t = a)$ to isolate regret contributions from each arm. This facilitates the integral form in the final step of the proof. Using the fact that the batch sizes are chosen to control for the regret to be balanced across batches, the idea is to construct an upper bound on the batch-wise arm specific regret, $R_a^{(m)}(\pi)$. Then, using Lemma 1, we can bound the expected cumulative regret. Note that unless otherwise stated, the covariate dimension $d$ is treated as fixed. Constants hidden in the asymptotic notation may depend on $d$ (potentially exponentially).

**Theorem 1.** *Under Assumptions 1–4, and with the batch sizes as defined in equation 10 in Section 4.1, the regret of the proposed BaNk-UCB algorithm ($\pi$) is bounded by,*

$$R_T(\pi) \lesssim |\mathcal{A}| M T^{\frac{1-\gamma}{1-\gamma^M}} (\ln T)^{\gamma}, \tag{13}$$

*where $\gamma = \frac{1+\alpha}{2+d}$.*

*Proof.* For $\epsilon > 0$, we split $R_a^{(m)}$ into two terms:

$$R_a^{(m)} = \int_{\mathcal{X}} (f_*(x) - f_a(x)) p_a^{(m)}(x) 1(f_*(x) - f_a(x) > \epsilon) dx$$

$$+ \int_{\mathcal{X}} (f_*(x) - f_a(x)) p_a^{(m)}(x) 1(f_*(x) - f_a(x) \le \epsilon) dx. \tag{14}$$

The idea is to bound these two terms separately, where the second one can be bounded using the margin assumption (i.e., Assumption 4). The $\epsilon$ is determined theoretically based on the bound on $R_a^{(m)}$. From Lemmas 8 and 10 in the Appendix B, we get that:

$$\int_{\mathcal{X}} (f_*(x) - f_a(x)) p_a^{(m)}(x) 1 (f_*(x) - f_a(x) > \epsilon) \, dx \lesssim \epsilon^{\alpha - d - 1} \ln t_{m-1} + t_m \epsilon^{1+\alpha}. \tag{15}$$

Furthermore, we can bound the second term in equation 14 by

$$\int_{\mathcal{X}} (f_*(x) - f_a(x)) p_a^{(m)}(x) 1 (f_*(x) - f_a(x) \le \epsilon) \, dx$$

$$\overset{(\dagger)}{\le} t_m \epsilon \int p_X(x) 1 (f_*(x) - f_a(x) \le \epsilon) \, dx$$

$$\overset{(\ddagger)}{\lesssim} t_m \epsilon^{1+\alpha}, \tag{16}$$

where $(\dagger)$ follows from Lemma 2 and $(\ddagger)$ follows from the Margin condition. Now combining equation 15 and equation 16, we get from equation 14:

$$R_a^{(m)} \lesssim \epsilon^{\alpha - d - 1} \ln t_{m-1} + t_m \epsilon^{1+\alpha} \tag{17}$$

By the choice of our batch end points $t_m = \lfloor a t_{m-1}^{\gamma} \rfloor$, then it is easy to see using a geometric sum in the exponent, $t_m = \Theta(T^{\frac{1-\gamma^m}{1-\gamma^M}})$ with $\gamma = \frac{1+\alpha}{2+d}$. Now, balancing the two terms in equation 17 and solving for $\epsilon$, we get $\epsilon = [t_m^{-1} \ln t_{m-1}]^{\frac{1}{2+d}}$. Therefore, we have:

$$R_a^{(m)} \lesssim t_m [t_{m-1}^{-1} \ln t_{m-1}]^{\frac{1+\alpha}{2+d}} \lesssim T^{\frac{1-\gamma^m}{1-\gamma^M}} \cdot T^{-\left(\frac{1-\gamma^{m-1}}{1-\gamma^M}\right)\left(\frac{1+\alpha}{2+d}\right)} \cdot (\ln t_{m-1})^{\frac{1+\alpha}{2+d}} = T^{\frac{1-\gamma}{1-\gamma^M}} (\ln t_{m-1})^{\gamma}. \tag{18}$$

Now, using Lemma 1,

$$R_T(\pi) = \sum_{a \in \mathcal{A}} \sum_{m=1}^{M} R_a^{(m)}(\pi)$$

$$\lesssim \sum_{a \in \mathcal{A}} \sum_{m=1}^{M} T^{\frac{1-\gamma}{1-\gamma^M}} (\ln t_{m-1})^{\gamma}$$

$$\lesssim |\mathcal{A}| M T^{\frac{1-\gamma}{1-\gamma^M}} (\ln T)^{\gamma}.$$

$\square$

Next, we state the minimax lower bound on the regret achievable by any M-batch policy $(\mathcal{G}, \pi)$ as established by Jiang & Ma (2025) and show that it matches the upper bound in Theorem 1 up to logarithm factors.

**Remark 3** (Comparison with Previous Work)**.** *Since Jiang & Ma (2025) is the only prior work that addresses the batched nonparametric bandit setting, it is important to emphasize that our proof techniques differ substantially from theirs. While their analysis builds on the binning-based framework of Perchet & Rigollet (2013), our regret analysis requires non-trivial extensions of the adaptive k-NN UCB algorithm of Zhao et al. (2024) to the batched setting. In particular, our analysis is fundamentally batch-aware: all supporting lemmas and the final regret bound are developed by carefully balancing the batch endpoints and are first established in a batch-wise fashion. Moreover, our supporting lemmas refine the analysis in Zhao et al. (2024) by clarifying implicit assumptions and extending the argument to handle the batch-constrained feedback setting. These technical developments are essential to handling the delayed feedback and restricted policy updates that characterize the batched regime.*

Our main contribution lies in achieving the same minimax-optimal regret rate as Jiang & Ma (2025), while introducing a conceptually simpler and data-adaptive algorithm that consistently outperforms binning-based methods in practice. In order to establish this, we present the fundamental limits of the batched nonparametric bandit problem as characterized by Jiang & Ma (2025).

**Theorem 2** (Minimax lower bound for nonparametric batched bandits; adapted from Jiang & Ma (2025))**.** *Let $\mathcal{F}(L, \alpha)$ denote the class of functions satisfying the Lipschitz smoothness condition (Assumption 3) with constant $L$ and the margin condition (Assumption 4). For any $M$-batch policy $\pi$ deployed over $T$ rounds in a 2-armed setting with reward functions $f_1, f_2 \in \mathcal{F}(L, \alpha)$, the minimax expected cumulative regret satisfies:*

$$\inf_{\pi} \sup_{f_1, f_2 \in \mathcal{F}(L,\alpha)} R_T(\pi) \gtrsim T^{\frac{1-\gamma}{1-\gamma^M}}, \quad \text{where } \gamma = \frac{\alpha+1}{2+d}.$$

This result characterizes the fundamental difficulty of learning in nonparametric batched bandits under the class $\mathcal{F}(L, \alpha)$, and shows that our BaNk-UCB algorithm matches this lower bound up to logarithmic factors. Note that, when $M \gtrsim \ln(\ln T)$ and the number of arms $|\mathcal{A}| \lesssim \ln T$, the cumulative regret simplifies to $R_T(\pi) = \tilde{O}(T^{1-\gamma})$, recovering the known minimax optimal rate for fully sequential (non-batched) nonparametric bandits (Perchet & Rigollet, 2013). This condition implies that, surprisingly, only a relatively modest increase in the number of batches (log-logarithmic in the horizon $T$) is sufficient to achieve the fully sequential optimal rate. Additionally, the mild logarithmic restriction on the number of actions $|\mathcal{A}|$ reflects practical scenarios where the action set is moderately large but not excessively growing with $T$, highlighting the efficiency of the BaNk-UCB algorithm in nearly matching fully adaptive performance despite batching constraints.

## 5 Experiments

In this section, we present numerical simulations and real-data experiments to illustrate the performance of the proposed Batched Nonparametric k-NN UCB algorithm (BaNk-UCB) in comparison to the nonparametric analogue: Batched Successive Elimination with Dynamic Binning (BaSEDB) algorithm of Jiang & Ma (2025).

### 5.1 Simulated Data

We evaluate `BaNk-UCB` under two simulation settings. Setting 1 deliberately violates the Lipschitz smoothness assumption and features a discontinuous, piecewise-constant reward structure to assess robustness under model misspecification. In contrast, Setting 2 satisfies the smoothness assumptions and serves as a benchmark under well-specified conditions. We describe both settings below.

**Setting 1:** We consider a piecewise-constant reward structure with localized high-reward regions, designed to challenge algorithmic adaptivity in non-smooth settings. Specifically, we define

$$f_1(x) = \sum_{j=1}^{D} v_j h \cdot 1\{x \in \mathcal{B}_j\}, \quad f_2(x) = 0, \quad x \in \mathcal{X},$$

where $v_j \in \{-1, 1\}$ are Rademacher variables and each $\mathcal{B}_j$ is a ball of radius $r$ centered at $c_j$. In Figure 1, we set $\mathcal{X} = [-1, 1]^d$ (with $d = 2$ and uniform $P_X$), $r = 0.6$, and $D = 6$, with randomly chosen centers and signs

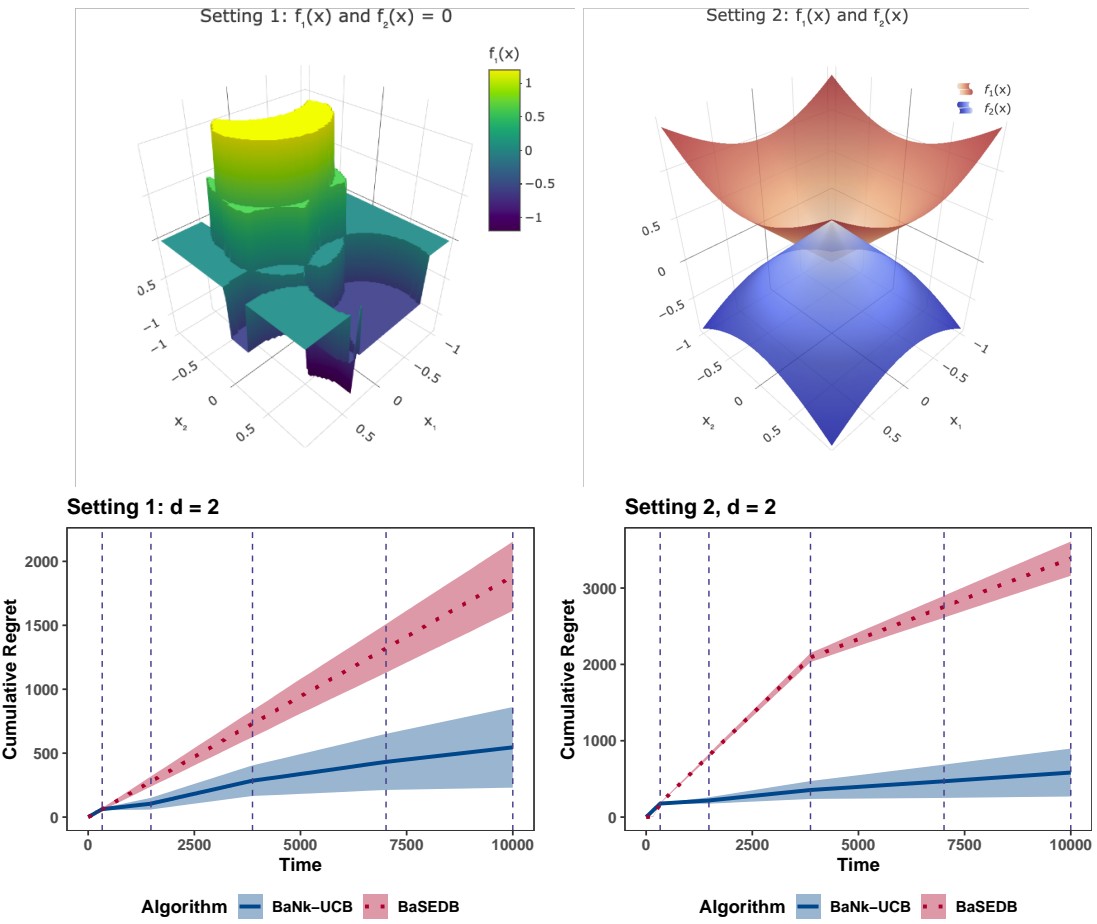

Figure 1: Top row (left to right): Reward functions for the two arms in Setting 1 and 2, respectively. Bottom row: Cumulative regret comparison for BaSEDB and BaNk-UCB algorithms over 30 runs.

for the balls. Similar function classes (typically in smoothed form) with sharp transitions and varying best arms across regions have appeared in related work, often as part of lower bound constructions in Perchet & Rigollet (2013), Jiang & Ma (2025), and Zhao et al. (2024). While this construction does not satisfy Lipschitz continuity, we include it to examine the robustness of `BaNk-UCB` under model misspecification.

**Setting 2:** As illustrated in Figure 1 consider the following choice of mean reward functions: $f_1(x) = \|x\|_2$ and $f_2(x) = 0.5 - \|x\|_2$, where $X$ is sampled uniformly from $[-1, 1]^d$, with $d = 2$.

We set $T = 10000$, $L = 1$ for the Lipschitz constant in Assumption 3. We fix the number of batches to $M = 5$ to balance between frequent updates and computational efficiency, but the results remain consistent across different choices of $M$. For the BaSEDB algorithm, we follow the specifications described in Jiang & Ma (2025) for choosing grid points and bin-widths. For our proposed BaNk-UCB algorithm, we choose the same batch grid for a fair comparison.

In Figure 1, we plot the cumulative regret averaged over 30 independent runs. In order to present an empirical assessment of the variability inherent in our simulations, the shaded regions represent empirical confidence intervals computed as $\pm 1$ times the standard error across these runs. The vertical dotted blue lines denote the grid choices for the batches.

`BaNk-UCB` consistently outperforms `BaSEDB` across all experimental settings. Although our batch sizes were selected based on empirical performance, they align closely with the theoretically motivated schedule in Section 4.1. Importantly, as illustrated in Appendix B.1, `BaNk-UCB` demonstrates robust performance across different batch schedules, provided the endpoints follow the prescribed growth pattern. This suggests that the algorithm does not require fine-tuned batch timing to perform effectively.

In Appendix B.1, we extend the comparison to higher-dimensional contexts ($d = 3, 4, 5$), where both methods degrade in performance, yet `BaNk-UCB` maintains a consistent advantage over `BaSEDB`. A key practical benefit of `BaNk-UCB` is its minimal tuning overhead. Unlike binning-based algorithms such as `BaSEDB`, which depend on careful calibration of bin widths, refinement rates, and arm elimination thresholds—often requiring knowledge of problem-specific parameters—`BaNk-UCB` relies on a fully data-driven nearest neighbor strategy. Its adaptively chosen $k$ automatically balances bias and variance based on local data density, without needing explicit smoothness or margin parameters. This makes `BaNk-UCB` both more robust to misspecification and easier to implement in practice.

### 5.2 Real Data

We evaluate the performance of BaNk-UCB and BaSEDB algorithm on three publicly available classification datasets: (a) *Rice* (Cammeo & Osmancik, 2020), consisting of 3810 samples with 7 morphological features used to classify two rice varieties; (b) *Occupancy Detection* (Candanedo & Feldheim, 2016), with 8143 samples and 5 environmental sensor features used to predict room occupancy; and (c) *EEG Eye State* (Biermann, 2014), with 14980 samples and 14 EEG measurements used to classify eye state. In all cases, we treat the

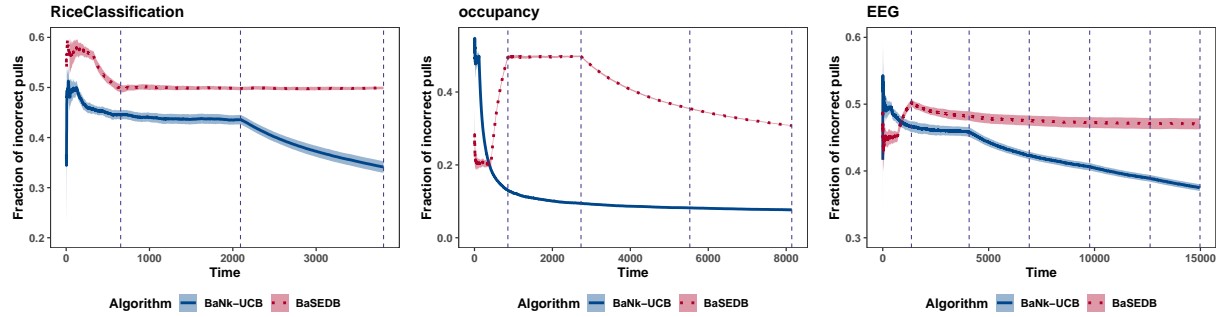

Figure 2: Rolling average fraction of incorrect decisions across three real datasets. BaNk-UCB achieves lower error and faster learning than BaSEDB.

true label as the optimal action and assign a binary reward of 1 if the selected action matches the label, and 0 otherwise. We simulate a contextual bandit setting where the context $x_t$ is observed, the learner selects an arm $a_t \in \{1, \dots, K\}$, and observes only the reward for the chosen arm. We set the number of arms $K$ equal to the number of classes (which is $K = 2$ for the three datasets considered) and choose 3, 4, and 6 batches respectively, aiming for sufficient granularity to allow adaptation without making individual batches too small. While the total number of batches $M$ reflects practical deployment constraints, the batch endpoints $\mathcal{G} = \{t_m\}_{m=1}^M$ are set to approximately follow the geometric schedule in Section 4.3, which balances regret across batches and aligns with our theoretical guarantees. The rolling fraction of incorrect decisions is computed using a windowed average over 30 independent random permutations of each dataset. In Figure 2, we plot the rolling fraction of incorrect decisions with shaded regions ($\pm 1.96\times$ standard errors) for uncertainty quantification as a function of the number of observed instances. `BaNk-UCB` consistently outperforms `BaSEDB` across all datasets. For the EEG dataset, which has the highest context dimensionality, `BaNk-UCB` exhibits *faster convergence and consistently lower error*, suggesting its advantage in *capturing local structure in high-dimensional spaces*. Batch sizes are chosen according to theoretical guidelines and are identical for both algorithms.

## 6 Conclusion

We introduced `BaNk-UCB`, a nonparametric algorithm for batched contextual bandits that combines adaptive $k$-nearest neighbor regression with the UCB principle. Unlike binning-based methods, `BaNk-UCB` leverages the local geometry of the context space and naturally adapts to heterogeneous data distributions. We established near-optimal regret guarantees under standard Lipschitz smoothness and margin conditions and proposed a theoretically grounded batch schedule that balances regret across batches. In addition to its theoretical robustness, empirically we illustrate that `BaNk-UCB` is resilient to batch scheduling choices and requires minimal parameter tuning, making it suitable for practical deployment in real-world systems. Empirical

evaluations on both synthetic and real-world classification datasets demonstrate that `BaNk-UCB` consistently outperforms existing nonparametric baselines, particularly in high-dimensional or irregular context spaces.

Despite these advantages, several open challenges remain. Although our regret guarantees show that $k$-NN performs well in moderate dimensions, its statistical accuracy may deteriorate in very high-dimensional regimes due to the regret bound's dependence on the ambient context dimension $d$. However, prior work on $k$-NN regression suggests that it can adapt to the intrinsic dimension of the context distribution, which may mitigate this issue. Formalizing this adaptation in the batched bandit setting remains an exciting direction for future work. Additionally, while our algorithm uses a theoretically motivated batch schedule, real-world systems may impose scheduling constraints that deviate from the idealized setting. Although our method performs well empirically under various batch schedules, deriving theoretical guarantees under arbitrary batch schedules is another important extension. Future work may also explore adaptive strategies for estimating smoothness and margin parameters, eliminating extraneous logarithmic factors in regret bounds, and generalizing the framework to infinite or structured action spaces.

### Broader Impact Statement

This work develops a theoretically grounded algorithm for sequential decision-making in batched settings, with applications in domains such as personalized medicine, online education, and adaptive experimentation. By improving statistical efficiency under limited feedback, our approach could contribute to safer and more effective decision-making in resource-constrained or high-stakes environments. However, care should be taken when applying such methods in sensitive domains, particularly in ensuring that fairness, transparency, and domain-specific constraints are accounted for. Our analysis does not directly consider fairness or robustness under distribution shift, and these remain important directions for future work.

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

## A Appendix

In this section, we provide the detailed proof for the regret upper bound for `BaNk-UCB` algorithm in Theorem 1. First we present the supporting lemmas for establishing the upper bound for the expected regret in Section B.

## B Proof for the Regret Upper Bound

Recall, the batch-wise expected sample density, $p_a^{(m)}(x)$, from equation 11. In Lemma 2, we first construct an upper bound for $p_a^{(m)}(x)$ in terms of the context density $p_X(x)$.

**Lemma 2.** *The batch-wise expected sample density satisfies:*

$$p_a^{(m)}(x) \leq (t_m - t_{m-1})p_X(x),$$

*for almost all $x \in \mathcal{X}$.*

*Proof.* Note, since the event $\{X_t \in A\} \subseteq \{X_t \in A, a_t = a\}$,

$$\mathbb{E}\left[\sum_{t=t_{m-1}}^{t_m} 1(X_t \in A, a_t = a)\right] \leq (t_m - t_{m-1})\int_A p_X(x)dx. \tag{19}$$

From equation 11 and equation 19, we get that,

$$\int_A p_a^{(m)}(x)dx \leq (t_m - t_{m-1})\int_A p_X(x)dx,$$

for all $A \in \mathcal{X}$. Therefore, $p_a^{(m)}(x) \leq (t_m - t_{m-1})p_X(x)$ for almost all $x \in \mathcal{X}$. $\square$

Next, we build a concentration bound on the average model noise for the $k$-nearest neighbors around a point $x$. Here, we will use the sub-Gaussianity of noise (Assumption 1) and the fact that we only observe data until the last batch, i.e., for $t \in [t_{m-1} + 1, t_m]$, we can only utilize data until time $t_{m-1}$ for estimation.

**Lemma 3.** *Let $N_{t_{m-1},k}(x,a)$ denote the set of $k$ nearest neighbors among $\{X_i : i < t_{m-1}, a_i = a\}$. Then, for all $x \in \mathcal{X}$, $a \in \mathcal{A}$, and $k \geq 1$, we have that,*

$$\mathbb{P}\left(\sup_{x,a,k}\left|\frac{1}{\sqrt{k}}\sum_{i \in \mathcal{N}_{t_{m-1},k}(x,a)}\epsilon_i\right| > u\right) \leq dt_{m-1}^{2d+1}|\mathcal{A}|e^{-\frac{u^2}{2\sigma^2}}, \tag{20}$$

*where $\epsilon_i$ are independent sub-Gaussian noise terms with variance proxy $\sigma^2$.*

*Proof.* From Lemma 4 of Zhao et al. (2024), we have that of a fixed $k$:

$$\mathbb{P}\left(\sup_{x,a}\left|\frac{1}{\sqrt{k}}\sum_{i \in \mathcal{N}_{t_{m-1},k}(x,a)}\epsilon_i\right| > u\right) \leq dt_{m-1}^{2d}|\mathcal{A}|e^{-\frac{u^2}{2\sigma^2}}. \tag{21}$$

Then we apply a union bound over all $k \leq t_{m-1}$ to get,

$$\mathbb{P}\left(\sup_{x,a,k}\left|\frac{1}{\sqrt{k}}\sum_{i \in \mathcal{N}_{t,k}(x,a)}\epsilon_i\right| > u\right) \leq dt_{m-1}^{2d+1}|\mathcal{A}|e^{-\frac{u^2}{2\sigma^2}}.$$

$\square$

Note, that Lemma 3 is for any batch $m$ and we will use it to bound the batch-wise regret.

**Definition 1.** *Define the event $\mathcal{E}_m$ as*

$$\mathcal{E}_m := \left\{\left|\frac{1}{\sqrt{k}}\sum_{i \in \mathcal{N}_{t_{m-1},k}(x,a)}\epsilon_i\right| \leq \sqrt{2\sigma^2\ln(dt_{m-1}^{2d+3}|\mathcal{A}|)} \,\,\forall\,\, x,a,k\right\}, \tag{22}$$

Then, from Lemma 3, it follows that $\mathbb{P}(\mathcal{E}_m) \geq 1 - 1/t_m$.

**Lemma 4.** *Under $\mathcal{E}_m$, we have that the following point-wise estimation error bound for $x \in \mathcal{X}$ and $t \in [t_{m-1}+1, t_m]$:*

$$f_a(x) \leq \widehat{f}_{a,t}(x) \leq f_a(x) + 2\xi_{a,t}(x) + 2Ld_{a,t}(x), \tag{23}$$

*where $\xi_{a,t}(x)$ and $d_{a,t}(x)$ are as defined in equation 8 and equation 5, respectively.*

*Proof.* Observe that for $t \in [t_{m-1}+1, t_m]$, under event $\mathcal{E}_m$ and $x \in \mathcal{X}$:

$$\left|\widehat{f}_{a,t}(x) - (f_a(x) + \xi_{a,t}(x) + Ld_{a,t}(x))\right| \tag{24}$$

$$\leq \left|\frac{1}{k_{a,t}(x)}\sum_{i \in \mathcal{N}_t(x,a)}(Y_i - f_a(x))\right|$$

$$\leq \frac{1}{k_{a,t}(x)}\sum_{i \in \mathcal{N}_t(x,a)}(Y_i - f_a(X_i)) + \frac{1}{k_{a,t}(x)}\sum_{i \in \mathcal{N}_t(x,a)}(f_a(X_i) - f_a(x))$$

$$\leq \xi_{a,t}(x) + Ld_{a,t}(x), \tag{25}$$

where the last line uses the definition of $\mathcal{E}_m$ in equation 22 and the Lipschitz (smoothness) property (Assumption 3) of $f_a$. $\square$

**Quantities of interest:** We define some important quantities of interest which are central to the proof. This includes two population quantities:

$$r_a(x) = \frac{1}{2L\sqrt{C_1}}(f_*(x) - f_a(x)), \tag{26}$$

$$n_a^{(m)}(x) = \frac{C_1 \ln t_{m-1}}{(f_*(x) - f_a(x))^2}, \tag{27}$$

in which

$$C_1 = \max\left\{4, 32\sigma^2(2d + 3 + \log(Md|\mathcal{A}|))\right\}. \tag{28}$$

The quantity $n_a^{(m)}(x)$ can be interpreted as a *local sample complexity proxy*, capturing the number of samples required near $x$ to estimate the reward function $f_a(x)$ with sufficient precision. Then, another quantity of interest is a data-dependent quantity that measures the total number of observations until time $t_{m-1}$ corresponding to arm $a$ in a radius $r$ ball around $x$. For any $x \in \mathcal{X}, a \in \mathcal{A}$ define,

$$n^{(m)}(x, a, r) := \sum_{t=1}^{t_{m-1}} 1(\|X_t - x\| < r, a_t = a). \tag{29}$$

Next in Lemma 5, under the event $\mathcal{E}_m$, we show that the adaptive choice of $k_{a,t}$ from equation 6 in our $k$-NN estimator is in fact upper bounded by $n_a^{(m)}(x)$. Then, in Lemma 6, we show that $n^{(m)}(x, a, r) \le k_{a,t}(x)$, which then leads to the relationship between $n_a^{(m)}(x)$ and $n^{(m)}(x, a, r)$ in Lemma 7.

**Lemma 5.** *Under event $\mathcal{E}_m$ for $t \in [t_{m-1} + 1, t_m]$,*

$$k_{a,t}(x) \le n_a^{(m)}(x).$$

*Proof.* We prove this by contradiction. Let $k_{a,t}(x) > n_a^{(m)}(x)$. By definition of $k_{a,t}$ in equation 6:

$$Ld_{a,t}(x) = Ld_{a,t,k_{a,t}(x)}(x) \le \sqrt{\frac{\ln(t_{m-1})}{k_{a,t}(x)}} \le \sqrt{\frac{\ln t_{m-1}}{n_a^{(m)}(x)}} = 2Lr_a(x), \tag{30}$$

From Lemma 4, under $\mathcal{E}_m$,

$$\widehat{f}_{a_t,t}(x) \le f_{a_t}(x) + 2\sqrt{\frac{2\sigma^2}{k_{a_t,t}(x)} \ln(dMt_{m-1}^{2d+3}|\mathcal{A}|)} + 2Lr_{a_t}(x)$$

$$\le f_{a_t}(x) + 2\sqrt{\frac{2\sigma^2}{n_{a_t}^{(m)}(x)} \ln(dMt_{m-1}^{2d+3}|\mathcal{A}|)} + 2Lr_{a_t}(x). \tag{31}$$

Since action $a_t$ is selected at time $t$, from the proposed UCB algorithm (Algorithm 1), i.e., the choice of $a_t = \arg\max_{a \in \mathcal{A}} \hat{f}_{a,t}(X_t)$ and from Lemma 4,

$$\hat{f}_{a_t,t}(x) \ge \hat{f}_{a^*(x),t}(x) \ge f_*(x). \tag{32}$$

Combining equation 31 and equation 32 gives:

$$2\sqrt{\frac{2\sigma^2}{n_{a_t}^{(m)}(x)} \ln(dt_{m-1}^{2d+3}|\mathcal{A}|)} + 2Lr_{a_t}(x) \ge f_*(x) - f_{a_t}(x). \tag{33}$$

We now derive an inequality that contradicts with equation 33. From equation 27 and equation 28,

$$2\sqrt{\frac{2\sigma^2}{n_{a_t}^{(m)}(x)} \ln(dt_{m-1}^{2d+3}|\mathcal{A}|)} = 2\sqrt{\frac{2\sigma^2}{C_1 \ln t_{m-1}} \ln(dt_{m-1}^{2d+3}|\mathcal{A}|)(f_*(x) - f_{a_t}(x))^2}$$

$$\le \frac{1}{2}\sqrt{\frac{\ln(dt_{m-1}^{2d+3}|\mathcal{A}|)}{(2d + 3 + \ln(d|\mathcal{A}|))\ln(t_{m-1})}}(f_*(x) - f_{a_t}(x))$$

$$< \frac{1}{2}(f_*(x) - f_{a_t}(x)). \tag{34}$$

From the definition of $r_a(x)$ in equation 26,

$$2Lr_{a_t}(x) = \frac{1}{\sqrt{C_1}}(f_*(x) - f_{a_t}(x)) \leq \frac{1}{2}(f_*(x) - f_{a_t}(x)). \tag{35}$$

From equation 34 and equation 35,

$$2\sqrt{\frac{2\sigma^2}{n_{a_t}^{(m)}(x)} \ln(dt_{m-1}^{2d+3}|\mathcal{A}|)} + 2Lr_{a_t}(x) < f_*(x) - f_{a_t}(x). \tag{36}$$

Note that equation 33 contradicts equation 36. Hence, the desired conclusion follows. $\qquad\square$

**Lemma 6.** *Under $\mathcal{E}_m$, let $r_a(x) \geq \frac{2LC_1}{\sqrt{C_1}-2}$ and $k_{a,t}(x) \gtrsim \ln T$, then, we get*

$$n^{(m)}(x, a, r_a(x)) \leq k_{a,t}(x),$$

*where $r_a(x)$ is as defined in equation 26, $n^{(m)}(x, a, r_a(x))$ defined in equation 29 and $k_{a,t}$ as defined in equation 6.*

*Proof.* We also prove Lemma 6 by contradiction. If $n^{(m)}(x, a, r_a(x)) > k_{a,t}(x)$, let

$$t = \max\{\tau < t_{m-1} \mid \|x_\tau - x\| \leq r_a(x), A_\tau = a\}. \tag{37}$$

be the last step falling in $B(x, r_a(x))$ with action $a$. Then $B(x, r_a(x)) \subseteq B(X_t, 2r_a(x))$, and thus there are at least $k_{a,t}(x)$ points in $B(X_t, 2r_a(x))$. Therefore, for any $x \in \mathcal{X}$, by the definition of $d_{a,t}(x)$, i.e., the distance of $x$ to its $k^{\text{th}}$ nearest-neighbors in equation 5,

$$d_{a,t}(x) < 2r_a(x). \tag{38}$$

Denote $a^*(x) = \arg\max_a f_a(x)$ as the best action at context $x$. Again, note that $a_t = a$ is selected only if the UCB of action $a$ is not less than the UCB of action $a^*(x)$, i.e.,

$$\hat{f}_{a,t}(X_t) \geq \hat{f}_{a^*(X_t),t}(X_t). \tag{39}$$

From Lemma 4,

$$\hat{f}_{a,t}(X_t) \leq f_a(X_t) + 2\xi_{a,t}(X_t) + 2Ld_{a,t}(X_t), \tag{40}$$

and

$$\hat{f}_{a^*(X_t),t}(X_t) \geq f_{a^*(X_t)}(X_t) = f_*(X_t). \tag{41}$$

From equation 39, equation 40, and equation 41,

$$f_a(X_t) + 2\xi_{a,t}(X_t) + 2Ld_{a,t}(X_t) \geq f_*(X_t). \tag{42}$$

which yields,

$$\begin{aligned}
d_{a,t}(X_t) &\geq \frac{f_*(X_t) - f_a(X_t) - 2\xi_{a,t}(X_t)}{2L} \\
&\geq \frac{f_*(X_t) - f_a(X_t) - 2\sqrt{\frac{2\sigma^2 \ln(dMT^{2d+3}|\mathcal{A}|)}{k_{a,t}(x)}}}{2L} \\
&\geq \frac{f_*(X_t) - f_a(X_t) - 2\sqrt{\frac{2\sigma^2 \ln(dMT^{2d+3}|\mathcal{A}|)}{\ln T}}}{2L} \\
&= \sqrt{C_1}r_a(X_t) - \frac{1}{L}\sqrt{\frac{2\sigma^2 \ln(dMT^{2d+3}|\mathcal{A}|)}{\ln T}} \\
&\geq \sqrt{C_1}r_a(X_t) - \frac{\sqrt{C_1}}{L} \\
&\geq 2r_a(X_t), \tag{43}
\end{aligned}$$

using the fact that $r_a(x) \geq \frac{2LC_1}{\sqrt{C_1}-2}$ and $k_{a,t}(x) \gtrsim \ln T$. Note that equation 43 contradicts equation 38. Therefore $n^{(m)}(x, a, r_a(x)) \leq k_{a,t}(x)$. That completes the proof of Lemma 6. $\qquad\square$

**Lemma 7.** *For $n_a(x)$ defined in equation 27 and $n^{(m)}(x, a, r)$ as defined in equation 29, under $\mathcal{E}_m$,*

$$n^{(m)}(x, a, r_a(x)) \le n_a^{(m)}(x).$$

*Proof.* Combining the results of Lemma 5 and 6 proves Lemma 7. [1]   □

**Bounding the batch-wise regret $R_a^{(m)}$:**   From Lemma 7 and from Lemma 3, we know that $\mathbb{P}(\mathcal{E}_m^c) \le 1/t_m$ and $n^{(m)}(x, a, r_a(x)) < t_m$ on $\mathcal{E}_m$ gives:

$$\mathbb{E}\left[n^{(m)}(x, a, r_a(x)) \mid \mathcal{F}_{t_{m-1}}\right] \le \mathbb{P}(\mathcal{E}_m|\mathcal{F}_{t_{m-1}})\mathbb{E}\left[n^{(m)}(x, a, r_a(x)) \mid \mathcal{E}_m, \mathcal{F}_{t_{m-1}}\right]$$
$$+ \mathbb{P}(\mathcal{E}_m^c|\mathcal{F}_{t_{m-1}})\mathbb{E}\left[n^{(m)}(x, a, r_a(x)) \mid \mathcal{E}_m^c, \mathcal{F}_{t_{m-1}}\right]$$
$$\le n_a^{(m)}(x) + 1. \tag{44}$$

From the definition of $p_a^{(m)}$ in equation 11,

$$\int_{B(x, r_a(x))} p_a^{(m)}(u) du \le n_a^{(m)}(x) + 1. \tag{45}$$

Recall $R_a^{(m)}$ from equation 12. We first bound $R_a^{(m)}$ for a given $m$ to get a bound on the expected regret using Lemma 1. To bound $R_a^{(m)}$, we introduce a new random variable $Z$ follow a distribution with probability density function (pdf) $\phi$:

$$\phi(z) = \frac{1}{C_Z\left[(f_*(z) - f_a(z)) \vee \epsilon\right]^d}, \tag{46}$$

where $C_Z$ is the normalizing constant. As discussed in Section 4, we split $R_a^{(m)}$ into two regions: one where the suboptimality gap is large (where concentration bounds dominate) and another where the margin condition helps control the measure of near-optimal points,

$$R_a^{(m)} = \int_{\mathcal{X}} (f_*(x) - f_a(x))p_a^{(m)}(x)1(f_*(x) - f_a(x) > \epsilon)dx$$
$$+ \int_{\mathcal{X}} (f_*(x) - f_a(x))p_a^{(m)}(x)1(f_*(x) - f_a(x) \le \epsilon)dx.$$

The idea is to bound these two terms separately, where the second one can be bounded using the margin assumption (i.e., Assumption 4). The $\epsilon$ is determined theoretically based on the bound on $R_a^{(m)}$. We tackle the first integral term in the following Lemma 8.

**Lemma 8.** *There exists a constant $C_2 > 0$ such that for any $a \in \mathcal{A}$,*

$$\int_{\mathcal{X}} (f_*(x) - f_a(x))p_a^{(m)}(x)1\left(f_*(x) - f_a(x) > \epsilon\right) dx$$
$$\le C_2 C_Z \, \mathbb{E}\left[\int_{B(Z, r_a(Z))} p_a^{(m)}(u)\left(f_*(u) - f_a(u)\right) du \,\middle|\, \mathcal{F}_{t_{m-1}}\right],$$

*where $Z \sim \phi$ is a density function defined over $\mathcal{X}$. Here $C_2$ is a constant that may depend exponentially on the covariate dimension d which we treat as fixed.*

---

[1]Lemmas 5 and 6 refine the argument used in Lemma 6 of Zhao et al. (2024), clarifying implicit assumptions and adapting the result to accommodate batched feedback.

*Proof.* Consider,

$$
\mathbb{E}\left[\int_{B(Z,r_a(Z))} p_a^{(m)}(u)\left(f_*(u)-f_a(u)\right)du \,\middle|\, \mathcal{F}_{t_{m-1}}\right]
\tag{47}
$$

$$
\overset{(a)}{=} \int_{\mathcal{X}}\int_{B(u,2r_a(u)/3)} \phi(z)p_a^{(m)}(u)\left(f_*(u)-f_a(u)\right)dzdu
$$

$$
\geq \int_{\mathcal{X}}\left(\inf_{\|z-u\|\leq 2r_a(u)/3}\phi(z)\right)\left(\frac{2}{3}\right)^d r_a^d(u)p_a^{(m)}(u)\left(f_*(u)-f_a(u)\right)du
$$

$$
\overset{(b)}{\geq} \left(\frac{2}{3}\right)^d\left(\frac{3}{4}\right)^d\int_{\mathcal{X}}\phi(u)r_a^d(u)p_a^{(m)}(u)\left(f_*(u)-f_a(u)\right)du
$$

$$
= \frac{1}{2^d C_Z}\int_{\mathcal{X}}\frac{1}{\left[(f_*(u)-f_a(u))\vee\epsilon\right]^d}r_a^d(u)p_a^{(m)}(u)\left(f_*(u)-f_a(u)\right)du
$$

$$
\geq \frac{1}{2^d C_Z}\int_{\mathcal{X}}1(f_*(u)-f_a(u)>\epsilon)\frac{1}{(f_*(u)-f_a(u))^d}\frac{(f_*(u)-f_a(u))^d}{(4L)^d}
$$

$$
\times p_a^{(m)}(u)\left(f_*(u)-f_a(u)\right)du
$$

$$
\geq \frac{1}{2^{3d}L^d C_Z}\int_{\mathcal{X}} p_a^{(m)}(u)\left(f_*(u)-f_a(u)\right)1(f_*(u)-f_a(u)>\epsilon)du.
\tag{48}
$$

For (a), if $\|u-z\|\leq r_a(z)$, then from the definition of $r_a$ in equation 26 and using the Lipschitz assumption (Assumption 3), we get that:

$$
\frac{r_a(u)}{r_a(z)} = \frac{f_*(u)-f_a(u)}{f_*(z)-f_a(z)}
$$

$$
= \frac{f_*(u)-f_*(z)+f_a(z)-f_a(u)+f_*(z)-f_a(z)}{f_*(z)-f_a(z)}
$$

$$
\leq \frac{f_*(z)-f_a(z)+2Lr_a(z)}{f_*(z)-f_a(z)}
$$

$$
= 1+\frac{1}{\sqrt{C_1}}
$$

$$
\leq \frac{3}{2}.
\tag{49}
$$

For (b), we have that $\|z-u\|\leq\frac{2r_a(u)}{3}$, therefore we have that:

$$
|f_*(u)-f_*(z)|\leq\frac{2}{3}r_a(u),\text{ and }|f_a(u)-f_a(z)|\leq\frac{2}{3}r_a(u).
$$

Therefore,

$$
|f_*(z)-f_a(z)-(f_*(u)-f_a(u))|\leq\frac{4}{3}r_a(u)
$$

$$
\Rightarrow (f_*(z)-f_a(z))\vee\epsilon\leq\left((f_*(u)-f_a(u)+\frac{4}{3}r_a(u))\right)\vee\epsilon.
$$

Therefore, we get that,

$$
\frac{\phi(z)}{\phi(u)} = \frac{\left[(f_*(u)-f_a(u))\vee\epsilon\right]^d}{\left[(f_*(z)-f_a(z))\vee\epsilon\right]^d}
$$

$$
\geq \frac{\left[(f_*(u)-f_a(u))\vee\epsilon\right]^d}{\left[(f_*(u)-f_a(u))+\frac{4}{3}Lr_a(u)\right]^d}
$$

$$
\geq \left(\frac{3}{4}\right)^d.
\tag{50}
$$

where equation 50 follows because,

$$f_*(u) - f_a(u) + \frac{4}{3}Lr_a(u) = f_*(u) - f_a(u) + \frac{4}{3}L \cdot \frac{1}{2L\sqrt{C_1}}(f_*(u) - f_a(u))$$

$$= (f_*(u) - f_a(u))\left(1 + \frac{2}{3\sqrt{C_1}}\right).$$

Since $\sqrt{C_1} \geq 2$, then equation 50 holds. $\square$

Next, we prove an inequality that plays a key role in bounding the regret contribution from contexts where the reward gap is large.

**Lemma 9.**

$$\int_{\mathcal{X}} (f_*(z) - f_a(z))^{-(d-1)} 1(f_*(z) - f_a(z) > \epsilon)\, dz \lesssim \begin{cases} \epsilon^{\alpha+1-d} & \text{if } d > \alpha + 1, \\ \log\left(\frac{1}{\epsilon}\right) & \text{if } d = \alpha + 1, \\ 1 & \text{if } d < \alpha + 1. \end{cases} \tag{51}$$

*Proof.* Consider

$$\int_{\mathcal{X}} (f_*(z) - f_a(z))^{-(d-1)} 1(f_*(z) - f_a(z) > \epsilon)\, dz \tag{52}$$

$$\overset{(a)}{\leq} \frac{1}{\underline{c}} \int_{\mathcal{X}} (f_*(z) - f_a(z))^{-(d-1)} 1(f_*(z) - f_a(z) > \epsilon)\, p_X(z)\, dz$$

$$\overset{(b)}{=} \frac{1}{\underline{c}} \mathbb{E}\left[(f_*(X) - f_a(X))^{-(d-1)} 1(f_*(X) - f_a(X) > \epsilon)\right]$$

$$= \frac{1}{\underline{c}} \int_0^\infty \mathbb{P}\left(\epsilon < f_*(X) - f_a(X) < t^{-\frac{1}{d-1}}\right) dt$$

$$\leq \frac{1}{\underline{c}} \int_0^{\epsilon^{-(d-1)}} \mathbb{P}\left(f_*(X) - f_a(X) < t^{-\frac{1}{d-1}}\right) dt \tag{53}$$

(a) comes from Assumption 2, which requires that $p_X(x) \geq \underline{c}$ over the support. In (b), the random variable $X$ follows a distribution with pdf $p_X$.

If $d > \alpha + 1$, then from Assumption 4,

$$\text{equation } 53 \leq \frac{D_\alpha}{\underline{c}} \int_0^{\epsilon^{-(d-1)}} t^{-\frac{\alpha}{d-1}}\, dt = \frac{D_\alpha(d-1)}{\underline{c}(d-1-\alpha)} \epsilon^{\alpha+1-d}. \tag{54}$$

If $d = \alpha + 1$, then

$$\text{equation } 53 \leq \frac{1}{\underline{c}} \int_0^1 dt + \frac{D_\alpha}{\underline{c}} \int_1^{\epsilon^{-(d-1)}} t^{-\frac{\alpha}{d-1}}\, dt = \frac{1}{\underline{c}} + \frac{D_\alpha(d-1)}{\underline{c}} \log\left(\frac{1}{\epsilon}\right). \tag{55}$$

If $d < \alpha + 1$, then

$$\text{equation } 53 \leq \frac{1}{\underline{c}} \int_0^1 dt + \frac{D_\alpha}{\underline{c}} \int_1^{\epsilon^{-(d-1)}} t^{-\frac{\alpha}{d-1}}\, dt \leq \frac{1}{\underline{c}} + \frac{D_\alpha(d-1)}{\underline{c}(\alpha+1-d)}. \tag{56}$$

Therefore, combining results from equation 53, equation 54, equation 55, and equation 56 we obtain:

$$\int_{\mathcal{X}} (f_*(z) - f_a(z))^{-(d-1)} 1(f_*(z) - f_a(z) > \epsilon)\, dz \lesssim \begin{cases} \frac{1}{\underline{c}}\epsilon^{\alpha+1-d} & \text{if } d > \alpha + 1, \\ \frac{1}{\underline{c}} \log\left(\frac{1}{\epsilon}\right) & \text{if } d = \alpha + 1, \\ \frac{1}{\underline{c}} & \text{if } d < \alpha + 1. \end{cases} \tag{57}$$

This proves equation 51. $\square$

**Lemma 10.** *Suppose Assumptions 1 and 2 hold. Then, for any batch $m \in [M]$, and for all arms $a \in \mathcal{A}$, we have:*

$$\mathbb{E}\left[\int_{B(Z,r_a(Z))} p_a^{(m)}(u)(\eta^*(u) - \eta_a(u))\,du \,\Big|\, \mathcal{F}_{t_{m-1}}\right] \lesssim \frac{1}{C_Z}\left(\epsilon^{\alpha-d-1}\log t_{m-1} + t_m\epsilon^{1+\alpha}\right).$$

*Here $C_Z$ is the density lower bound constant from equation 46 and $\mathcal{F}_{t_{m-1}}$ is the history until the $(m-1)^{th}$ batch.*

*Proof.* Consider:

$$\mathbb{E}\left[\int_{B(Z,r_a(Z))} p_a^{(m)}(u)(f_*(u) - f_a(u))\,du \,\Big|\, \mathcal{F}_{t_{m-1}}\right]$$

$$\overset{(a)}{\leq} \frac{3}{2}\mathbb{E}\left[\int_{B(Z,r_a(Z))} p_a^{(m)}(u)(f_*(z) - f_a(z))\,du \,\Big|\, \mathcal{F}_{t_{m-1}}\right]$$

$$\overset{(b)}{\leq} \frac{3}{2}\mathbb{E}\left[((n_a^{(m)}(Z) + 1) \wedge (t_m p_Z(z) r_a^d(Z)))(f_*(Z) - f_a(Z)) \,\Big|\, \mathcal{F}_{t_{m-1}}\right]$$

$$= \frac{3}{2}\int \left((n_a^{(m)}(z) + 1) \wedge (t_m p_Z(z) r_a^d(Z))\right)(f_*(z) - f_a(z))\frac{1}{\phi_Z[(f_*(z) - f_a(z)) \vee \epsilon]^d}dz$$

$$= \frac{3}{2}\int \left((n_a^{(m)}(z) + 1) \wedge (t_m p_Z(z) r_a^d(Z))\right)(f_*(z) - f_a(z))\frac{1}{\phi_Z[(f_*(z) - f_a(z))]^d}$$

$$\times 1(f_*(z) - f_a(z) > \epsilon)dz$$

$$+ \frac{3}{2}\int \left((n_a^{(m)}(z) + 1) \wedge (t_m p_Z(z) r_a^d(Z))\right)(f_*(z) - f_a(z))\frac{1}{\phi_Z\epsilon^d}1(f_*(z) - f_a(z) \leq \epsilon)dz, \quad (58)$$

For (a):

$$f_*(u) - f_a(u) \leq f_*(z) - f_a(z) + 2Lr_a(z)$$

$$\leq f_*(z) - f_a(z) + \frac{1}{\sqrt{C_1}}(f_*(z) - f_a(z))$$

$$\leq \frac{3}{2}(f_*(z) - f_a(z)). \quad (59)$$

We get (b) from Lemma 2 and equation 44. In equation 58, we split the domain based on whether $(f_*(z)-f_a(z))$ is large or small, and use the margin assumption (Assumption 4) for the latter. Note that, If $f_*(Z)-f_a(Z) > \epsilon$, then $n_a^{(m)}(Z) = (\log t_{m-1})(f_*(Z) - f_a(Z))^{-2}$ is smaller, otherwise the bias dominates.

$$\text{equation } 58 = \frac{3}{2C_Z}\left(\int \left(\frac{C_1 \ln t_{m-1}}{(f_*(z) - f_a(z))} + f_*(z) - f_a(z)\right)\frac{1}{(f_*(z) - f_a(z))^d}1(f_*(z) - f_a(z) > \epsilon)dz\right.$$

$$\left. + \int t_m p_Z(z) r_a^d(Z)(f_*(z) - f_a(z))\frac{1}{\epsilon^d}1(f_*(z) - f_a(z) \leq \epsilon)dz\right)$$

$$\lesssim \frac{1}{C_Z}\left(\mathbb{E}\left[(f_*(Z) - f_a(Z))^{-(d+1)}1(f_*(Z) - f_a(Z) > \epsilon)\right]\ln t_{m-1}\right.$$

$$\left. + \frac{t_m}{\epsilon^d}\mathbb{E}\left[(f_*(Z) - f_a(Z))^{d+1}1(f_*(Z) - f_a(Z) \leq \epsilon)\right]\right)$$

$$\overset{(c)}{\lesssim} \frac{1}{C_Z}\left(\epsilon^{\alpha-d-1}\ln t_{m-1} + t_m\epsilon^{1+\alpha}\right),$$

where the first term in (c) comes from the dominating term in Lemma 9 and for the second term we use the Margin assumption as follows:

$$\int_{\mathcal{X}}(f_*(z) - f_a(z))1(f_*(z) - f_a(z) < \epsilon)dz \leq \frac{1}{\underline{c}}\mathbb{E}\left[(f_*(X) - f_a(X))1(f_*(X) - f_a(X) < \epsilon)\right]$$

$$\leq \frac{L_0}{\underline{c}}\epsilon^{\alpha+1}. \quad (60)$$

This concludes the proof. □

## B.1 Additional Experiments in Higher Dimensions

We extend the numerical experiments from Section 5.1 to evaluate algorithm performance in higher-dimensional contexts. Specifically, we consider $d \in \{3, 4, 5\}$ while keeping the underlying data-generating mechanisms for both experimental settings unchanged. As expected, the performance of both `BaSEDB` and `BaNk-UCB` deteriorates with increasing dimension, consistent with the theoretical prediction from Theorem 1 and Theorem 2 that regret decays more slowly when $d$ is large due to the corresponding decrease in the parameter $\gamma$.

Despite the increased difficulty, `BaNk-UCB` continues to outperform `BaSEDB` across all settings, including the more challenging Setting 1. These results highlight the robustness of `BaNk-UCB` in moderate to high-dimensional settings, where the benefits of adapting to local geometry become even more pronounced.

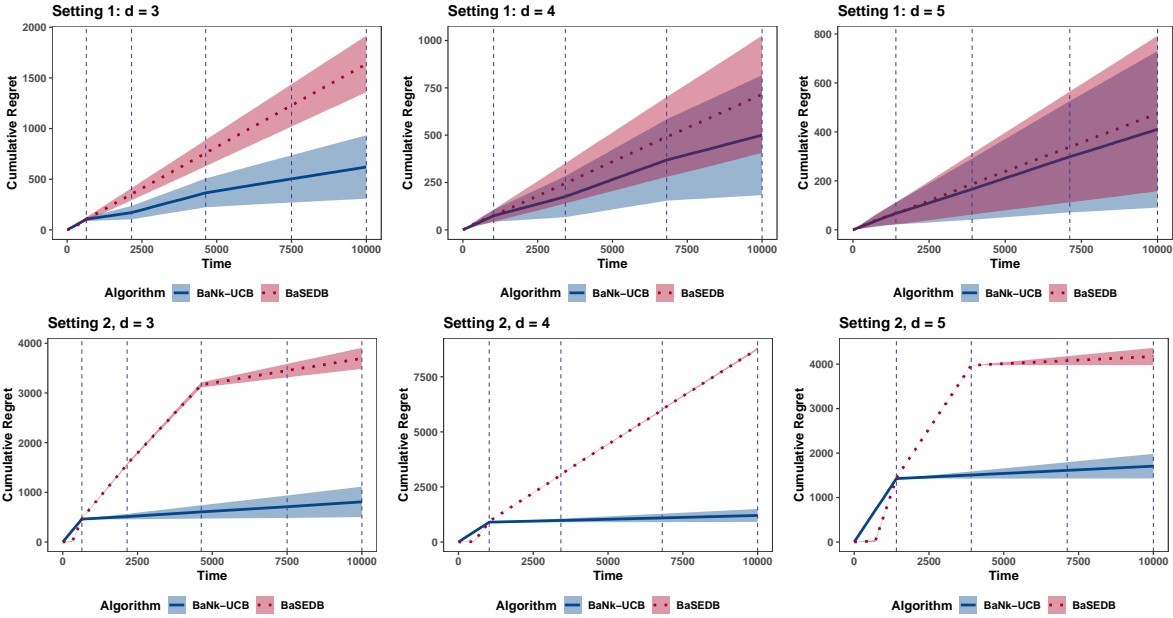

Figure 3: Average cumulative regret over 30 runs for `BaSEDB` and `BaNk-UCB` under Settings 1 and 2 with $d \in \{3, 4, 5\}$. Vertical dashed lines denote batch boundaries.

## Appendix D: Batch Scheduling Strategies for Robustness Evaluation

To assess the robustness of `BaNk-UCB` to the choice of batch timing, we evaluate its performance under three distinct scheduling strategies. Each scheme partitions the total time horizon $T$ into $M$ non-overlapping batches of varying lengths, subject to the constraint $\sum_{m=1}^{M} t_m = T$.

1. **Uniform Schedule.** All batches have equal length:

$$t_m = \left\lfloor \frac{T}{M} \right\rfloor, \quad \text{for } m = 1, \dots, M.$$

This baseline distributes samples evenly across time.

2. **Linearly Increasing Schedule.** Batch lengths grow linearly with index:

$$t_m = \left\lfloor \frac{m}{\sum_{j=1}^{M} j} \cdot T \right\rfloor, \quad \text{for } m = 1, \dots, M.$$

This emphasizes early exploration and later exploitation.

3. **Normalized Exponential Schedule.** Batch sizes grow exponentially:

$$w_m = 1 - \texttt{base}^{-m}, \quad \text{with } \texttt{base} > 1,$$

$$t_m = \left\lfloor \frac{w_m}{\sum_{j=1}^{M} w_j} \cdot T \right\rfloor.$$

This accelerates sample allocation in later stages; we set $\texttt{base} = 2$ in our experiments.

The resulting breakpoints $\{t_1, t_1 + t_2, \ldots, T\}$ determine when policy updates occur in the batched setting.

**Empirical Findings.** Figure 4 compares cumulative regret under all three scheduling schemes for both simulation settings. `BaNk-UCB` consistently achieves lower regret than `BaSEDB`, regardless of batch timing strategy. These results underscore the robustness of `BaNk-UCB` to the choice of batch schedule.

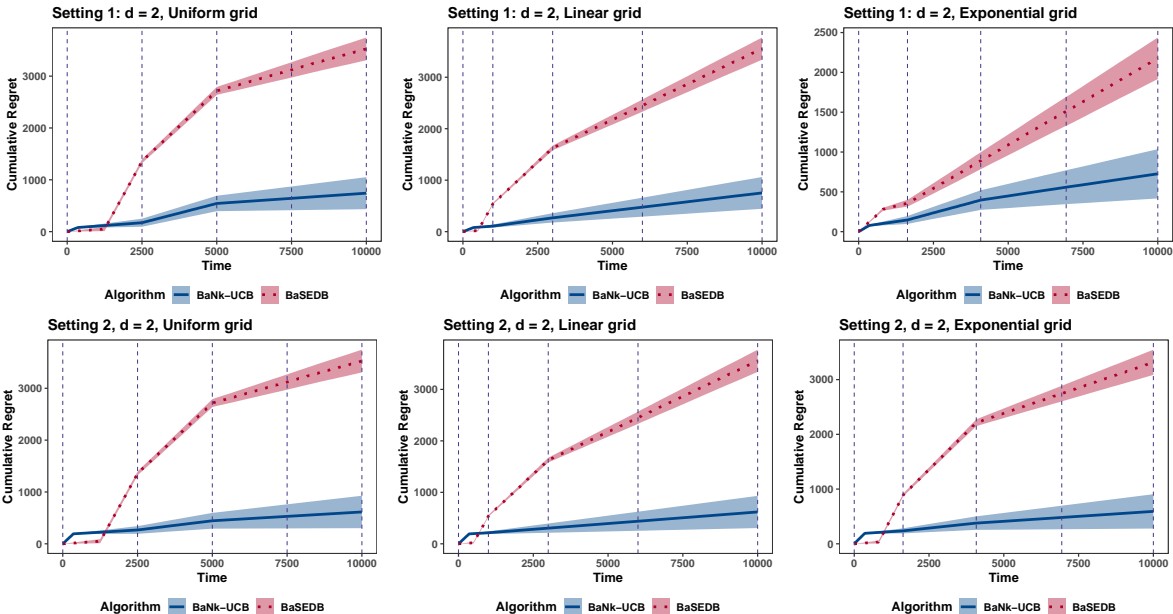

Figure 4: Average cumulative regret over 30 runs for `BaSEDB` and `BaNk-UCB` under Settings 1 and 2 with $d = 2$ across three batch scheduling strategies. Vertical dashed lines denote the corresponding batch boundaries.

