# OpenReview forum: "Batched Nonparametric Bandits via k-Nearest Neighbor UCB"
_TMLR — Accepted by TMLR_

### Review · Reviewer_bLYY · 2025-08-25

**Summary Of Contributions:**

The paper proposes a nonparametric approach to the batched contextual bandit problem. The upper confidence bound (UCB) is evaluated through the adaptive k-nearest neighbor (k-NN) regression, in which the number of neighbors k can be adaptively determined. The authors shows regret guarantees under standard assumptions such as Lipschitz smoothness and margin assumption. Empirical evaluation is provided for synthetic and real-world datasets.

**Audience:**

Yes

**Audience Explanation:**

k-NN is easy to use in practice, and so, providing a theoretical justification should be informative.

**Claims And Evidence:**

No

**Claims Explanation:**

Strengths:

- The problem setting, batched nonparametric contextual bandit, is a topic worth investigating.

- Detailed theoretical analysis for the regret bound is provided (though I couldn't follow the entire proof).

Weaknesses:

- I currently do not fully understand if the relation with an existing study is property described.

- Several technical details are not clear to me, as questioned below.

**Requested Changes:**

I think that the connection with (Zhao et al., 2024) is stronger than what is stated in the main text. While the Intro and Related Work sections do not emphasize this relationship, the proposed method essentially appears to be a batch extension of (Zhao et al., 2024). If that is the case, it should be clearly stated in the Intro and Related Work in more detail. For example, the adaptive k-NN seems directly from (Zhao et al., 2024), but the way it is described in the Intro could lead to the misunderstanding that it is original to this paper.

In the experiments, what is the idea behind the settings of \cal{G} and t_m? I do not fully understand typical settings of t_m and M. Are they given by the problem setting or can they be determined by the user? In the later case, under the what criterion, the user selects them? Are they kind of tuning parameters?

In Sec 5.2, the authors mentioned 'The number of batches was selected based on the total number of samples to ensure reasonable granularity while maintaining batch sizes that approximately align with our theoretically motivated geometric schedule'. What is 'reasonable granularity'? What does 'approximately align with our theoretically motivated geometric schedule' mean?

Although the authors claim that the required parameter tuning of the proposed method is minimal, the following (possibly) tuning issues are not clear to me.
- In practice, how is L chosen? Further, while L = 1 in simulation data, what value was used in real-data and how is that value justified?
- How is \sigma in (8) determined?

For setting 1,
- What does description 'Setting 1 is derived from the regret lower bound construction' mean?
- Setting 1 does not satisfy the Lipschitz smoothness assumption. What is the purpose for using a function that does not meet required assumption in the first experiment of the paper?

The authors claim that it is empirically shown the proposed method is resilient wrt scheduling. However, in my understanding, a result with only one scheduling is shown for each one of settings. To claim robustness wrt scheduling, different results with different schedulings should be shown for one fixed setting.

In the right bottom plot of Figure 1, the blue line breaks different locations from the dashed vertical line while other results seemingly break at the dashed vertical lines. Is it correct?

Why do the lines in Figure 2 change between vertical dashed lines. In my current understanding, the lines here should be piece-wise constant if we have sufficient amount of results (although it seems moving average, the window size of the average is small (30), so I guessed the result should become almost piece-wise constant. Do I misunderstand something?).

---

> ### Author Response · Authors · 2025-09-15
> **Response to Reviewer bLYY: Part 1**
>
> Dear Reviewer bLYY, Thank you for your constructive feedback and insightful questions. Below are our detailed response:
>
> 1. *I think that the connection with *Zhao et al, 2025* is stronger than what is stated in the main text. While the Intro and Related Work sections do not emphasize this relationship, the proposed method essentially appears to be a batch extension of *(Zhao et al, 2025)* [1]. If that is the case, it should be clearly stated in the Intro and Related Work in more detail. For example, the adaptive k-NN seems directly from [1], but the way it is described in the Intro could lead to the misunderstanding that it is original to this paper.*
>
> We appreciate the reviewer's close reading and the opportunity to clarify the relationship between our work and *Zhao et al, 2025*. Indeed, our algorithm adopts the adaptive $k$-NN framework introduced there for online contextual bandits, and extends it to the batched setting with delayed feedback. This connection is stated explicitly in the BaNk-UCB section and a technical remark, but we agree it deserves clearer emphasis in the Introduction and Related Work to avoid any impression that the adaptive $k$-NN idea originated with our paper.
>
> At the same time, we would like to clarify that the broader line of work on nonparametric $k$-NN bandits precedes [1]. For instance, [2] and [3] explore $k$-NN-based strategies in both full-information and bandit settings. Our contribution builds on this broader literature and adapts these ideas to the *batched* contextual bandit setting, which raises distinct technical challenges not addressed in prior work. For instance Lemmas 5 and 6 in our paper refine the concentration arguments of [1] by making implicit assumptions explicit, incorporating batchwise index bounds, and handling the restricted policy updates inherent to the batched regime.
>
> To address the reviewer’s concern, we have revised the Introduction and Related Work to: (a) acknowledge [1] as the immediate precursor to our adaptive $k$-NN UCB construction; (b) situate that within the broader nonparametric bandit literature; and (c) highlight the specific batch-aware advances made in this work. We hope this addresses the concern.
>
>
> 2. *In the experiments, what is the idea behind the settings of $\cal{G}$ and $t_m$? I do not fully understand typical settings of $t_m$ and $M$. Are they given by the problem setting or can they be determined by the user? In the later case, under what criterion, the user selects them? Are they kind of tuning parameters?*
>
> In our experiments, the number of batches $M$ and the corresponding batch endpoints $\mathcal{G} = \\{t_m\\}_{m=1}^M$ are user-specified and reflect the level of adaptivity allowed by the application. Typically, \\(M\\) is determined by practical constraints such as budget cycles, trial stages, or deployment logistics, that limit how often policies can be updated.
>
> The batch endpoints \\(\\{t_m\\}\\), while user-specified, are informed by theoretical considerations. Specifically, their construction depends on problem-specific parameters such as the margin parameter \\(\\alpha\\) and context dimension \\(d\\), and is designed to balance regret across batches. To this end, we adopt a geometrically increasing schedule:
> \\[
> t_1 = ad, \\quad t_m = \\lfloor a t_{m-1}^\\gamma \\rfloor, \\quad \\text{where } \\gamma = \\frac{1 + \\alpha}{2 + d},
> \\]
> with \\(a = \\Theta\\big(T^{\\frac{1 - \\gamma}{1 - \\gamma^M}}\\big)\\) calibrated so that \\(t_M = T\\). This ensures that later batches gather enough data for accurate reward estimation and achieves minimax-optimal regret under our assumptions.
>
> We also clarify that \\(\\mathcal{G} = \\{t_m\\}_{m=1}^M\\) denotes the batch endpoints, and \\(t_0 = 0\\) is used by convention to mark the start of the first batch.
>
> **References**
>
> [1] Zhao, P., Fan, R., Wang, S., Shen, L., Zhang, Q., ZongKe, and Zheng, T. (2025). Contextual bandits for
> unbounded context distributions. In Forty-second International Conference on Machine Learning.
>
> [2] Guan, M. and Jiang, H. (2018). Nonparametric stochastic contextual bandits. In Proceedings of the AAAI
> Conference on Artificial Intelligence, volume 32.
>
> [3] Reeve, H., Mellor, J., and Brown, G. (2018). The k-nearest neighbour ucb algorithm for multi-armed bandits
> with covariates. In Algorithmic Learning Theory, pages 725–752. PMLR.

---

> ### Author Response · Authors · 2025-09-15
> **Response to Reviewer bLYY: Part 2**
>
> 3. *In Sec 5.2, the authors mentioned 'The number of batches was selected based on the total number of samples to ensure reasonable granularity while maintaining batch sizes that approximately align with our theoretically motivated geometric schedule'. What is 'reasonable granularity'? What does 'approximately align with our theoretically motivated geometric schedule' mean?*
>
>
> We thank the reviewer for this helpful question. We have revised the text in Section 5.2 to clarify this point. Specifically, we now write:
>
> ``We set the number of arms $K$ equal to the number of classes (which is $K = 2$ for the three datasets considered) and choose 3, 4, and 6 batches respectively, aiming for sufficient granularity to allow adaptation without making individual batches too small. While the total number of batches \\(M\\) reflects practical deployment constraints, the batch endpoints \\(\\mathcal{G} = \\{t_m\\}_{m=1}^M\\) are set to approximately follow the geometric schedule in Section 4.3, which balances regret across batches and aligns with our theoretical guarantees.''
>
> Here, ``reasonable granularity'' refers to a tradeoff between adaptivity and statistical stability: using more batches improves adaptivity but risks insufficient data per batch. The geometric schedule provides a principled way to space out batch endpoints to ensure that each batch receives sufficient samples for reliable estimation while achieving near-optimal regret rates. We hope this revision clarifies the motivation behind our batch design.
>
> 4. *Although the authors claim that the required parameter tuning of the proposed method is minimal, the following (possibly) tuning issues are not clear to me.*
>
>   *1. In practice, how is $L$ chosen? Further, while $L = 1$ in simulation data, what value was used in real-data and how is that value justified?*
>
>    *2.  How is $\sigma$ in (8) determined?*
>
> 1. In both simulation and real-data experiments, we set $L = 1$. This choice is motivated by the fact that we normalize all contexts (e.g., scaling components to $[0,1]$ or unit variance), and the rewards are bounded in the simulations and binary in nature in real data. As a result, the underlying reward functions $f_a(x) $ are bounded and exhibit smooth behavior. Under these conditions, the Lipschitz constant is finite, and a conservative value such as $L = 1$ provides both theoretical validity and computational simplicity. In more complex or less controlled settings, $L$ could be selected via grid search or cross-validation.
>
> 2. Since rewards $r_t \\in \\{0,1\\}$ are bounded in our real-data settings, they are also sub-Gaussian with variance proxy $\\sigma^2 \\leq 1/4$. We set $\\sigma^2 = 1/4$ in all experiments to reflect this property and to ensure valid confidence bounds in the UCB strategy. This is a standard assumption in the bandit literature for bounded binary rewards.
>
> *5) For setting 1,*
>
> *1. What does description 'Setting 1 is derived from the regret lower bound construction' mean?*
>
> *2. Setting 1 does not satisfy the Lipschitz smoothness assumption. What is the purpose for using a function that does not meet required assumption in the first experiment of the paper?*
>
> We thank the reviewer for their thoughtful questions regarding the design and motivation behind Setting 1.
>
> 1. While our function is piecewise constant and does not satisfy the global Lipschitz condition, similar function classes (typically in smoothed form) with sharp transitions and varying best arms across regions have appeared in related work, often as part of lower bound constructions in [4], [5], and [1].  While this construction does not satisfy Lipschitz continuity, we include it to examine the robustness of BaNk-UCB under model misspecification.
>
> 2. To address this, we now clarify in the revised manuscript that our experiments are intentionally designed to assess performance both when the assumptions hold and when they are violated. Specifically, we include two settings:
>
> - **Setting 1:** A piecewise constant function that violates the Lipschitz smoothness assumption, allowing us to evaluate whether BaNk-UCB remains effective even under model misspecification.
> - **Setting 2:** A reward structure with $f_1(x) =\\|x\\|_2$ and $f_2(x) = 0.5 - \\|x\\|_2$, which satisfies the Lipschitz condition with constant $L=1$.
>
> This contrast highlights both the theoretical grounding and empirical robustness of our algorithm.
>
>
> **References contd.**
>
> [4] Perchet, V. and Rigollet, P. (2013). The multi-armed bandit problem with covariates. The Annals of Statistics.
>
> [5] Jiang, R. and Ma, C. (2025). Batched nonparametric contextual bandits. IEEE Transactions on Information
> Theory.

---

> ### Author Response · Authors · 2025-09-15
> **Response to Reviewer bLYY: Part 3**
>
> 6. *The authors claim that it is empirically shown the proposed method is resilient w.r.t. scheduling. However, in my understanding, a result with only one scheduling is shown for each one of settings. To claim robustness wrt scheduling, different results with different schedulings should be shown for one fixed setting.*
>
> In the original manuscript, we used a theoretically motivated geometric batch schedule (based on the smoothness and margin parameters) for each setting. However, we agree that demonstrating robustness to scheduling requires more than a single instantiation.
> In response, we have now added additional experiments in both Settings 1 and  2, where we fix the data-generating process and vary the batch schedule as follows:
> 1. **Uniform Schedule:** Batches of equal size, i.e., split the total horizon $T$ into $M$ intervals of length $T/M$.
> 2. **Linear Schedule:** Batch lengths increase linearly across time, ensuring more samples in later stages.
> 3. **Normalized Exponential Schedule:**  Batch sizes follow a truncated exponential growth pattern and these are parameter agnostic unlike the geometric schedule proposed in  the paper.
>
> Our updated results (included in Appendix D) show that BaNk-UCB performs better than BaSEDB across these schedules, with only minor differences in early-stage regret due to granularity. This supports our claim that the algorithm is robust to batch scheduling, provided that the schedule offers sufficient coverage and growth.
>
> 7. *In the right bottom plot of Figure 1, the blue line breaks different locations from the dashed vertical line while other results seemingly break at the dashed vertical lines. Is it correct?*
>
> Yes. In general, we do expect shifts in the regret trajectory near the dashed vertical lines, which denote batch endpoints. This is because in the batched bandit setting, feedback from all actions taken in a batch is revealed only at the end of that batch, allowing the learner to update its policy. Consequently, regret curves often flatten or improve after batch boundaries.
>
> However, for methods like our proposed $k$-NN UCB, which are locally adaptive and rely on neighborhood-specific statistics, the impact of these updates can manifest more gradually or at slightly different locations. In particular, if the updated confidence intervals lead to significantly improved arm selection in a specific region of the context space, the regret may drop sharply even before or after the batch endpoint. Such behavior is less pronounced in binning-based methods, where updates are globally tied to fixed spatial partitions and tend to align more cleanly with batch boundaries.
>
>
> 8. *Why do the lines in Figure 2 change between vertical dashed lines. In my current understanding, the lines here should be piece-wise constant if we have sufficient amount of results (although it seems moving average, the window size of the average is small (30), so I guessed the result should become almost piece-wise constant. Do I misunderstand something?).*
>
> We thank the reviewer for this insightful question. Indeed, in Figure 2, the regret curves are smoothed using a moving average with window size 30, which introduces some continuity across batch endpoints. However, even without smoothing, the regret incurred within each batch is not necessarily piecewise constant, especially for nonparametric methods like BaNk-UCB that adapt locally and may achieve improved arm selection over time within a batch.
>
> Specifically, while the policy itself is only updated at batch boundaries, the cumulative regret continues to accrue throughout each batch. If the policy selects better actions more frequently (e.g., due to better neighborhood estimates in high-density regions), the slope of the cumulative regret curve may decrease mid-batch, leading to curvature or non-constant behavior. This phenomenon is particularly visible in adaptive methods like ours, which leverage the geometry of the context space to guide exploration and can outperform static methods even without mid-batch updates.

---

### Review · Reviewer_7TbF · 2025-08-31

**Summary Of Contributions:**

This paper studies decision-making in batched contextual bandits under limited feedback settings. It proposed a new method, called BaNk-UCB, that combines adaptive k-NN regression with the UCB principle to guide actions without parametric assumptions. It adapts to context density, achieves near-optimal regret guarantees, and balances exploration and exploitation across batches. Experiments show BaNk-UCB outperforms binning-based baselines in synthetic and read datasets.

**Audience:**

Yes

**Audience Explanation:**

The batched nonparametric bandit is a relatively new setting.  This paper proposes a new method that connects k-nearest neighbor analysis with the classical UCB algorithm. This could be interesting to researchers of the bandit community.

**Broader Impact Concerns:**

The paper has included a well-written broader impact statement. I think it has been well addressed.

**Claims And Evidence:**

Yes

**Claims Explanation:**

The overall presentation is good, and the theoretical proof can support the claims. However, here are some issues to be resolved.

1. (A minor issue about the problem setting) In the regret definition Equation (3), $X_t$ is a random variable. Do you think there should be an expectation over $X_t$?

2. It’s better to include a detailed discussion of Remark 1, instead of directly referring to some previous works.

3. In equation 8, it requires the preknowledge of the subGaussian variance $\sigma^2$. This assumption is unusual, and do you think it can be relaxed? I’m also curious about the $\log (t^{2d+3})$ term. Could you provide an intuitive explanation for it?


4. In the proof of Lemma 1, first, there are some typos. The $f_{a_t}$ should be $f_a$ in the third and fourth equations. Second, the notation here is weird, as the right side of equation 12 does not depend on $\pi$. I guess it’s because the expectation on the left is taken over both $X_t$ and $a_t$, and $a_t$ depends on $\pi$ (maybe $\pi_m$. This notation on dependence should be revised to be clearer.

5. Proof of Theorem 1: Typos: $f^*$ should be $f_ *$ in equation (15)(16).

6. Presentation of Theorem 2: There is a $\sup_{f_1,f_2}$. However, the objective does not explicitly depend on $f_1,f_2$. The notation should be further polished. A similar issue occurs in the presentation of experiments. It should be made clear what $f_1,f_2$ are. I guess it represents that the action set contains only 2 elements, and $f_1,f_2$ are the reward functions for them, is it correct?

**Requested Changes:**

See the Weaknesses part. I think most issues could be fixed, but the current notations can be somewhat confusing.

---

> ### Author Response · Authors · 2025-09-15
> **Response to Reviewer 7TbF-- Part 1**
>
> Dear reviewer 7TbF,
> Thank you for your helpful feedback and insightful questions. Please find our detailed
> answers below:
>
> 1. *(A minor issue about the problem setting) In the regret definition Equation (3), $X_t$ is a random variable. Do you think there should be an expectation over $X_t$?*
>
> Just above Equation (3), we define the expected cumulative regret as $\mathcal{R}_T(\pi) = \mathbb{E}[R_T(\pi)]$, where the italicized $\mathcal{R}_T$ explicitly denotes the expected regret. In Equation (3), $R_T(\pi)$ refers to the random cumulative regret, and the expectation is taken over the randomness in the contexts $X_t$ and actions $a_t$ under policy $\pi$. We will highlight this distinction more clearly in the revision to avoid confusion.
>
> 2. *It’s better to include a detailed discussion of Remark 1, instead of directly referring to some previous works.*
>
> We have revised Remark 1 to provide a more detailed explanation. Specifically, *(Rigollet and Zeevi, 2013)* [3] showed that under the smoothness condition with exponent $\beta$, if the margin exponent $\alpha$ satisfies $\alpha\beta > d$, then the oracle policy becomes trivial, i.e., it always selects the same arm, regardless of the context. This implies that the context carries no useful information for the decision process. Since our work assumes Lipschitz smoothness ($\beta = 1$), we restrict attention to $\alpha \leq d$ to ensure the context remains relevant. The revised remark now states this clearly.
>
> 3. *In equation 8, it requires the preknowledge of the subGaussian variance $\sigma^2$. This assumption is unusual, and do you think it can be relaxed? I’m also curious about the $\log (t^{2d+3})$ term. Could you provide an intuitive explanation for it?*
>
>  While the assumption of a known sub-Gaussian variance proxy $\sigma^2$ in Equation (8) may appear restrictive, it is a standard analytical device in the nonparametric bandits literature. For example, *(Reeve et al, 2018)*[2] assume unit-variance sub-Gaussian noise for simplicity, and earlier works such as *(Perchet and Rigollet, 2013)*[3] assume bounded rewards, which also imply sub-Gaussian tails (with variance proxy at most $1/4$ under bounded support in $[0,1]$). We make this dependence explicit by allowing a general $\sigma^2$, so the scaling in our UCB widths and regret remains transparent.
> In practice,  $\sigma^2$ can be estimated or heuristically specified, by 1) conservatively setting to a large value, 2) treated as a tuning parameter, or, 3) estimated as the empirical variance of residuals from $k$-NN fits in early batches.
>
> The term $\log(t_{m-1}^{2d + 3})$ arises from a high-probability concentration bound on the average sub-Gaussian noise terms computed from data observed up to batch $m$ as formalized in Lemma 3. Specifically, the exponent $2d + 3$ reflects a union bound taken over:
> (i) a $t_{m-1}^d$-sized cover of the context space (in both the query point and neighbors, yielding $t_{m-1}^{2d}$ combinations),
> (ii) all arms $a \in \mathcal{A}$, and
> (iii) all neighborhood sizes $k \leq t_{m-1}$.
> This guarantees uniform high-probability bounds on the deviations of the empirical reward estimates from their conditional expectations, uniformly over all $(x, a, k)$ combinations considered up to batch $m$, with failure probability at most $1/t_m$.
>
> 4. *In the proof of Lemma 1, first, there are some typos. The $f_{a_t}$ should be $f_a$ in the third and fourth equations. Second, the notation here is weird, as the right side of equation 12 does not depend on $\pi$. I guess it’s because the expectation on the left is taken over both $X_t$ and $a_t$, and $a_t$ depends on $\pi$ (maybe $\pi_m$). This notation on dependence should be revised to be clearer.*
>
> We would like to clarify that there is no inconsistency: the third line in the proof transitions to a summation over arms $a \in \mathcal{A}$, not over the observed arm $a_t$. Specifically, the indicator $\mathbf{1}(a_t = a)$ ensures that we only accumulate regret terms corresponding to times when arm $a$ was selected. This trick allows us to express the expectation over $a_t \sim \pi(\cdot \mid X_t)$ as a sum over actions, facilitating a clean integral representation in the final step. To avoid confusion, we have now revised the notation slightly and corrected the variable in the final integral (replacing $X_t$ with the dummy variable $x$). We also added a clarifying remark in the proof to emphasize that the summation over $a \in \mathcal{A}$ is indexing actions taken, not introducing a new random variable.
>
>
> **References**
>
> [1] *Rigollet, P. and Zeevi, A. (2010). Nonparametric bandits with covariates. Conference on Learning Theory
> (COLT), page 54.*
>
> [2]*Reeve, H., Mellor, J., and Brown, G. (2018). The k-nearest neighbour ucb algorithm for multi-armed bandits
> with covariates. In Algorithmic Learning Theory, pages 725–752. PMLR.*
>
> [3]*Perchet, V. and Rigollet, P. (2013). The multi-armed bandit problem with covariates. The Annals of Statistics.*

---

> ### Author Response · Authors · 2025-09-15
> **Response to reviewer 7TbF- Part 2**
>
> 5. *Proof of Theorem 1: Typos: $f^\star$ should be $f_\star$ in equation $(15)(16)$.*
>
> We have corrected all instances of $f^\star$ to the intended $f_\star$ in Equations (15) and (16) and elsewhere.
>
> 6. *Presentation of Theorem 2: There is a $\sup_{f_1,f_2}$. However, the objective does not explicitly depend on $f_1,f_2$. The notation should be further polished. A similar issue occurs in the presentation of experiments. It should be made clear what $f_1,f_2$ are. I guess it represents that the action set contains only 2 elements, and $f_1,f_2$ are the reward functions for them, is it correct?*
>
> We thank the reviewer for pointing out this notational ambiguity. Indeed, the lower bound in Theorem 2 considers the worst-case regret over all 2-arm instances where each reward function $f_1, f_2$ belongs to the nonparametric class $\mathcal{F}(L, \alpha)$. We agree that this was not clearly stated and may be confusing. We have now clarified in the manuscript that the action set is binary ($|\mathcal{A}| = 2$), and $f_1, f_2$ correspond to the reward functions of the two arms. We have also revised the theorem statement and surrounding text accordingly to improve readability.

---

### Review · Reviewer_tmYZ · 2025-09-23

**Summary Of Contributions:**

This paper studies the nonparametric contextual bandit problem under a batch constraint. The authors propose an algorithm that combines k-nearest neighbor regression with the UCB idea, and establish an upper bound on its regret. The effectiveness of the proposed algorithm is also validated by numerical experiments.

**Audience:**

Yes

**Audience Explanation:**

Overall, the paper proposes a novel approach to a natural problem setting, and I believe it will attract attention from the machine learning community for both its applications and techniques. That said, I have a couple of further concerns:

* The paper assumes that the batch size can be chosen by the learner. In what applications is such an assumption justified?
* The batch size proposed in Equation (10) leads to very unbalanced batch sizes. This seems unrealistic for the motivating applications and practical constraints described in the Introduction. Could the authors address this concern? (I note that similar issues could also apply to other batched bandit approaches.)

I would appreciate the authors’ responses to the above questions.

**Claims And Evidence:**

Yes

**Claims Explanation:**

From my reading, all theoretical claims are either proven in the paper or supported by references, and I did not find any obvious mistakes. The numerical experiments also appear sufficiently reproducible and provide solid support for the claims. However, some parts of the appendix could be better organized.

I do have a few questions regarding the technical details:

* **Assumption 4 vs. prior work**: Assumption 4 seems to differ from the setting in Rigollet & Zeevi (2010). Is this assumption a natural extension, and is the description in Remark 1 accurate?
  In the prior work, the assumption appears to correspond to the probability that, for $a \neq a'$, we have $0 < |f\_a(X) - f\_{a'}(X)| \le \delta$.

* **On Assumption 4 and Remark 1**: Why is it that $\alpha \le d$ in Assumption 4, while in Remark 1 we have $\alpha \le 1$?

* **Theorem 1 regret bound**: In the special case $M=T$, the batch constraint disappears and the problem reduces to the standard setting, which should be easier. However, Theorem 1’s upper bound then becomes $O(T)$, which seems somewhat counterintuitive. Could the authors clarify this point?

* **Lemma 8 constant $C\_2$**: According to the proof, the constant \$C\_2\$ grows exponentially with the dimension \$d\$. Is it still acceptable to call this a “constant”? If the analysis treats \$d\$ as fixed, it would be better to state this explicitly. Note that this exponential dependence also appears in the upper bound of Theorem 1.

**Requested Changes:**

I would appreciate the authors’ responses to the above questions.

**Minor comments**:

* Please make the notation of the indicator function consistent (use either $\mathbf{1}$ or $1$ throughout).
* In some places where the proof follows immediately after a lemma (e.g., Lemma 3, Lemma 6), the heading still reads *“Proof of Lemma \*\*”*. This could be simplified for consistency.

---

> ### Author Response · Authors · 2025-09-27
> **Response to Reviewer tmYZ: Part 1**
>
> Dear reviewer tmYZ, Thank you for your helpful feedback and insightful questions. Please find our detailed explanations below:
>
> 1. **Assumption 4 vs. prior work:** *Assumption 4 seems to differ from the setting in Rigollet \& Zeevi (2010). Is this assumption a natural extension, and is the description in Remark 1 accurate? In the prior work, the assumption appears to correspond to the probability that, for $a \neq a'$, we have
>     \\[
>     0 < |f_a(X) - f_{a'}(X)| \leq \delta.
>   \\]*
>
> We thank the reviewer for raising this point.
> In Rigollet and Zeevi (2010), which studies the two-arm case, the margin condition is stated as
> \\[
> P_X \big[0 < |f^{(1)}(X) - f^{(2)}(X)| \leq \delta \big] \ \leq\  C \delta^\alpha,
> \qquad \forall \delta \in [0,\delta_0].
> \\]
> This condition controls the probability mass of contexts where the two arms are nearly indistinguishable.
>
> In our multi-arm setting, we extend this idea by requiring the margin condition to hold for the suboptimality gap relative to the best arm:
> \\[
> P_X\\big[0 < f^*(X) - f_a(X) \leq \delta \big]  \leq\ C \delta^\alpha,
> \qquad \forall a \neq a^*.
> \\]
> When $K=2$, this reduces to their two-armed case, since the only suboptimal arm is the runner-up. For $K > 2$, our condition generalizes the same principle to each suboptimal arm individually.
> We have clarified this connection after the assumption statement.
>
> 2. **On Assumption 4 and Remark 1:** *Why is it that $\alpha \leq d$ in Assumption 4, while in Remark 1 we have $\alpha \leq 1$?*
>
> The conditions $\alpha \leq d$ in Assumption 4 and $\alpha \leq 1$ in Remark 1 reflect different but compatible results. Perchet & Rigollet (2013) show via regret lower bounds that $\alpha\beta \leq d$ is the sharp, dimension-dependent threshold, and  when $\alpha\beta>d$, the oracle is trivial. This motivates Assumption 4. By contrast, Tsybakov and Audibert (2007), as noted in Rigollet and Zeevi (2010), show that $\alpha(1\wedge\beta)>1$ is a dimension-free sufficient condition for triviality, in which case the Bayes classifier is constant and contexts cease to matter. Thus, $\alpha \leq d$ governs the general bandit setting, while $\alpha \leq 1$ reflects a stronger sufficient condition from classification. To avoid conflating the two, we now cite only the bandit-specific result in Remark 1.
>
> 3. **Theorem 1 regret bound:**  *In the special case $\alpha = \infty$, the batch constraint disappears and the problem reduces to the standard setting, which should be easier. However, Theorem 1’s upper bound then becomes $\mathcal{O}(T)$, which seems somewhat counterintuitive. Could the authors clarify this point?*
>
> Our theorems explicitly focus on the nontrivial, finite-margin regime $\alpha \leq d$ (cf. Assumption 4 and Theorem 1).
> In this regime, the regret analysis balances an estimation term with a margin term controlled by $\alpha$, which leads to the exponent $\gamma=(1+\alpha)/(2+d)$ and the rate in Theorem 1.
>
> When $\alpha= \infty$ (hard margin), the problem changes qualitatively: there exists a uniform gap
> $\Delta =\inf_x\min_{a\neq a^*(x)} (f^\star(x)-f_a(x)) >0$. The "near-boundary" set has zero mass, so the
> second term in the standard decomposition of batch-wise regret (eqn (14)),
> \\[
> \int (f^\star(x)-f_a(x)) p_a^{(m)}(x) 1(f^\star(x)-f_a(x)\le\delta)  dx,
> \\]
> vanishes for any $\delta<\Delta$. Consequently, no balancing step in $\delta$ is needed and the finite-$\alpha$ proof template (which produces the $\gamma$ term) is not the right tool. The problem reduces to best-arm identification with a fixed gap. Using the same concentration lemmas for the $k$-NN estimates plus a UCB-style counting argument, one obtains a gap-dependent logarithmic bound. Thus, directly plugging $\alpha=\infty$
> into the finite-$\alpha$ bound yields a vacuous $\mathcal O(T)$ rate; the correct analysis for the hard-margin case gives $\tilde{\mathcal O}(\log T)$ regret and batching is no longer rate-limiting.
>
> 4. **Lemma 8 constant $C_2$:**  *According to the proof, the constant $C_2$ grows exponentially with the dimension $d$. Is it still acceptable to call this a “constant”? If the analysis treats $d$ as fixed, it would be better to state this explicitly. Note that this exponential dependence also appears in the upper bound of Theorem 1.*
>
> Yes, $C_2$ depends exponentially on $d$ through covering number arguments, reflecting the curse of dimensionality. Like Perchet and Rigollet (2013), Jiang and Ma (2025), and Zhao et al. (2025), we treat $d$ as fixed, so $C_2$ is a legitimate constant. We now clarify this in the revised manuscript.

---

> ### Author Response · Authors · 2025-09-27
> **Response to Reviewer tmYZ: Part 2**
>
> **Further concerns**
>
> 1. *The paper assumes that the batch size can be chosen by the learner. In what applications is such an assumption justified?*
>
> In practice, batch sizes are not chosen arbitrarily, but reflect operational constraints in many applications. For example: (i) in clinical trials, policies are updated only after each trial phase or cohort; (ii) in agriculture, recommendations are revised once per growing stage (e.g., planting, mid-season, harvest); (iii) in online platforms, algorithms are retrained periodically due to budget or deployment cycles. Our formulation follows prior work on batched bandits (Perchet et al. (2016), Jiang and Ma (2025)), where the learner is allowed to set batch endpoints $\{t_m\}$ subject to a total budget $T$.  Finally, as shown in Appendix D, our experiments with uniform, linear, and exponential schedules demonstrate that BaNk-UCB is empirically robust to different batching choices, suggesting that its performance does not hinge on any particular schedule.
>
> 2. *The batch size proposed in Equation (10) leads to very unbalanced batch sizes. This seems unrealistic for the motivating applications and practical constraints described in the Introduction. Could the authors address this concern? (I note that similar issues could also apply to other batched bandit approaches.*
>
> We thank the reviewer for this insightful point. Indeed, the geometrically increasing batch schedule in Equation (10) can lead to highly unbalanced batch sizes. This choice is not meant as a literal prescription for practice, but as a theoretically motivated abstraction that equalizes the regret contribution across batches and achieves minimax-optimal rates under the smoothness and margin assumptions. In real applications, batch sizes are typically application-driven. To address this, Appendix D reports results under uniform, linear, and exponential schedules, where *BaNk-UCB* consistently outperforms *BaSEDB*, demonstrating robustness.
> Developing theory for application-driven batching is an interesting direction for future work.
>
> **Minor Comments:**
> We have standardized the notation for the indicator function throughout the manuscript (using $1{(\cdot)}$ consistently). In addition, we have simplified the lemma proof headings (e.g., “Proof of Lemma 3” $\to$ “Proof”) for consistency across Lemmas 3, 6, and others. These corrections have been implemented in the revised version.
>
> **References**
> - Tsybakov, A. and Audibert, J.-Y. (2007). Fast learning rates for plug-in classifiers. Annals of Statistics,
> 35(2):608–633.
> - Rigollet, P. and Zeevi, A. (2010). Nonparametric bandits with covariates. Conference on Learning Theory
> (COLT), page 54.
> - Perchet, V. and Rigollet, P. (2013). The multi-armed bandit problem with covariates. The Annals of Statistics.
> - Jiang, R. and Ma, C. (2025). Batched nonparametric contextual bandits. IEEE Transactions on Information
> Theory.
> - Zhao, P., Fan, R., Wang, S., Shen, L., Zhang, Q., ZongKe, and Zheng, T. (2025). Contextual bandits for
> unbounded context distributions. In Forty-second International Conference on Machine Learning.
> - Perchet, V., Rigollet, P., Chassang, S., and Snowberg, E. (2016). Batched bandit problems. The Annals of
> Statistics, 44(2):660 – 681.

---

### Comment · Reviewer_5hyd · 2025-08-14
**Request to reject reviewing this paper.**

Hi Editor,

Another editor reached out to me earlier and said he will assign a paper to me. I agreed but you assigned this paper before him. I'm working full-time now and have very limited time. I would like to reject reviewing this paper and take the other one as I promised. I hope you can understand.

Best,
Joey

---

### Decision · Action_Editor_6Ekr · 2025-11-05

**Recommendation:** Accept as is

**Additional Comments:**

I recommend you adjust the plots so the confidence areas do not go below 0 regret.

**Audience:**

Yes

**Audience Explanation:**

While the results rely heavily on the methods from prior work, the setting here is not treated in existing literature and hence could present interest to the community.

**Claims And Evidence:**

Yes

**Claims Explanation:**

The paper studies the setting of nonparametric batched contextual bandits where the learner has the ability to choose the batch schedule. The paper proposes a new method, BaNk-UCB, that combines adaptive k-NN regression with the UCB principle to guide actions without parametric assumptions and adapting to context density. The main claims of the paper are that this algorithm achieves near-optimal regret guarantees, this result being formally proved. Experiments show BaNk-UCB outperforms binning-based baselines in synthetic and 3 publicly available datasets. All reviewers are in agreement that the main claims of the paper are supported by satisfactory evidence.